# YAP inhibition enhances the differentiation of functional stem cell-derived insulin-producing β cells

Edwin A. Rosado-Olivieri [1], Kendall Anderson[1], Jennifer H. Kenty [1] & Douglas A. Melton [1]

Stem cell-derived insulin-producing beta cells (SC-β) offer an inexhaustible supply of functional β cells for cell replacement therapies and disease modeling for diabetes. While successful directed differentiation protocols for this cell type have been described, the mechanisms controlling its differentiation and function are not fully understood. Here we report that the Hippo pathway controls the proliferation and specification of pancreatic progenitors into the endocrine lineage. Downregulation of YAP, an effector of the pathway, enhances endocrine progenitor differentiation and the generation of SC-β cells with improved insulin secretion. A chemical inhibitor of YAP acts as an inducer of endocrine differentiation and reduces the presence of proliferative progenitor cells. Conversely, sustained activation of YAP results in impaired differentiation, blunted glucose-stimulated insulin secretion, and increased proliferation of SC-β cells. Together these results support a role for YAP in controlling the self-renewal and differentiation balance of pancreatic progenitors and limiting endocrine differentiation in vitro.

[1] Department of Stem Cell and Regenerative Biology, Harvard Stem Cell Institute, Harvard University, Cambridge, MA 02138, USA. Correspondence and requests for materials should be addressed to D.A.M. (email: dmelton@harvard.edu)

β cell loss is a hallmark of type I and type II diabetes, and cell replacement strategies have been explored to restore functional β cells[1,2]. Recently, approaches to direct the differentiation of hPSCs into endocrine cells have been demonstrated[3,4], providing an alternate source of β cells for cell replacement therapies, drug discovery, and disease modeling. While these protocols are based on developmental signals involved in in vivo pancreatic development, our understanding of how these signaling factors coordinate the last steps of β-cell differentiation is incomplete[5,6].

During pancreatic development, endocrine cells differentiate from multipotent pancreatic progenitors (MPPs) that express NGN3, a factor essential for endocrine specification[7–10]. Similar to what occurs during in vivo organogenesis, treatment with EGFs and thyroid hormone T3, along with BMP, TGF-β, and Notch inhibition, helps drive stem cell-derived pancreatic progenitors into NGN3-expressing endocrine progenitors[3,4]. Cell cycle arrest of these progenitors accompanies their further differentiation to β cells[8,11–13].

The in vitro-differentiated β cells express NKX6.1, PDX1, and insulin, among other genes, all of which are key to their glucose-stimulated insulin secretion (GSIS) function, an essential part of controlling glucose homeostasis in vivo[3,4,14,15]. Genetic studies have indicated a prominent role for NKX6.1 in the development of β cells from endocrine progenitors[14], and methods to enhance the numbers of pancreatic progenitors that express NKX6.1 from hPSCs have been described[3,4,16–19]. It is the subsequent step of differentiation, wherein pancreatic progenitors form monohormal β cells, that the signals controlling the differentiation are less well understood. The present study shows that YAP, a member of the Hippo signaling pathway, is involved in controlling the generation of functional β cells from MPPs.

The Hippo pathway has been shown to integrate tissue architecture by balancing progenitor cell self-renewal and differentiation[20]. Inhibition of Hippo signaling results in the nuclear translocation of the downstream effectors YAP and TAZ, which, upon binding to TEAD coactivators, regulate expression of genes involved in progenitor cell proliferation[20,21]. In contrast, sustained activation of the pathway by growth-restrictive signals promotes terminal differentiation of mature cell types by inducing the phosphorylation, cytoplasmic retention, and degradation of YAP/TAZ[21]. Constitutive activation of YAP/TAZ in the mouse pancreas results in reduced organ size, acute pancreatitis, and impaired endocrine differentiation[22,23]. YAP plays a role in the control of progenitor expansion and maintenance of human fetal and stem cell-derived MPPs by regulating enhancer elements of transcription factors involved in these processes[24]. A recent study showed that mechanotransduction controls YAP activity in MPPs to direct cell fate via integrin signaling[25]. Moreover, the downregulation of YAP has been documented in NGN3 + endocrine progenitors and islet cells[22–25]. However, the extensive loss of tissue architecture as a result of genetic perturbations of the pathway in vivo confounded an analysis of whether or how YAP controls differentiation in pancreatic endocrine lineages.

Taking advantage of the in vitro differentiation of SC-β cells, we ascribe a role for YAP as a regulator of progenitor self-renewal and differentiation. Our studies show that YAP regulates the self-renewal of early progenitors and formation of NKX6.1 + pancreatic progenitors. We further show that both the chemical and genetic downregulation of YAP enhance endocrine differentiation and the terminal differentiation of functional monohormonal β cells. Finally, we demonstrate the utility of a YAP inhibitor for the depletion of progenitor cells in vitro.

## Results

**YAP is downregulated during endocrine differentiation.** YAP expression was examined during the multistep directed differentiation of hPSCs into β cells as outlined in Fig. 1a[3]. We observed YAP protein expression throughout stages 3–6 (Fig. 1b–f and Supplementary Fig. 1a–c), including in PDX1 + early and NKX6.1 + late MPPs at stages 3 and 4 of differentiation, respectively (Fig. 1b, c). YAP downregulation begins late in stage 4 NKX6.1 + MPPs and is correlated with the expression of the pan-endocrine marker CHGA (Fig. 1c, f, g and Supplementary Fig. 1a–d). Although cytoplasmic and nuclear YAP expression is present in NKX6.1 + cells at stage 4, YAP expression in this subpopulation of MPPs further declines as differentiation proceeds into the endocrine lineage (Fig. 1g and Supplementary Fig. 1b, f).

Because differentiation of MPPs is not synchronous, both early (NGN3 +) and late (CHGA +) endocrine progenitors are present at stage 5, day 3. Immunostaining of cell clusters at this time point shows nuclear and cytoplasmic expression of YAP in ~30% of early endocrine progenitors (NGN3 +; Supplementary Fig. 1e, g), whereas a high proportion of late endocrine cells (CHGA +) have lost YAP expression (Fig. 1h and Supplementary Fig. 1c, e). Reduced YAP expression in endocrine cells persists upon completion of the directed differentiation protocol: ~95% of CHGA + endocrine and insulin-expressing β cells do not express YAP (Fig. 1e, f, h and Supplementary Fig. 1c). At these latter stages of differentiation, YAP expression is largely restricted to non-endocrine cells that co-express the ductal marker SOX9 + (Fig. 1i and Supplementary Fig. 1h, i). In all, YAP is expressed during the progenitor stages and its downregulation correlates with the commitment of MPPs into the endocrine and then beta cell lineages (Fig. 1f).

**YAP is necessary for the specification of late MPPs.** We sought to determine whether YAP regulates the balance between progenitor self-renewal and differentiation by treating cells with the YAP inhibitor verteporfin. This small molecule inhibits the interaction between YAP and TEAD coactivators and downstream target gene expression[26]. When verteporfin is added during the specification of early PDX1 + to late NKX6.1 + MPPs (Fig. 2a), there is a significant decrease in the number of NKX6.1 + progenitors compared with control differentiations, without any detectable effects on cell viability (Fig. 2b–d). The treatment also results in a twofold decrease in the proportion of MPPs coexpressing the proliferation marker Ki67 (Fig. 2e–g), consistent with the previous analysis[24]. In agreement with this result, YAP-targeted shRNAs expressed in early pancreatic progenitors during this stage of differentiation (Fig. 2h) results in a twofold downregulation of YAP at the RNA and protein level (Fig. 2i–j) and dramatically reduces the proportion of NKX6.1 + progenitors with respect to control differentiations (1.1 ± 0.8 vs. 41.4 ± 1.2%; Fig. 2k–m). Concomitantly, we observed a premature differentiation of MPPs into CHGA + /NKX6.1− endocrine cells in cultures of progenitors expressing YAP shRNAs (Fig. 2n). These results are consistent with a role for YAP in promoting the proliferation and expression of NKX6.1 + in MPPs and in restraining their differentiation into the endocrine lineage.

**YAP inhibition enhances the differentiation of β cells.** Since YAP activity is correlated with the self-renewal of progenitor cells[20,21] and its downregulation correlates with their commitment to the endocrine lineage (Fig. 1d–i, Supplementary Fig. 1a–c), we hypothesized that YAP downregulation might direct the differentiation of MPPs into the endocrine and β-cell lineages. When YAP was inhibited by verteporfin during

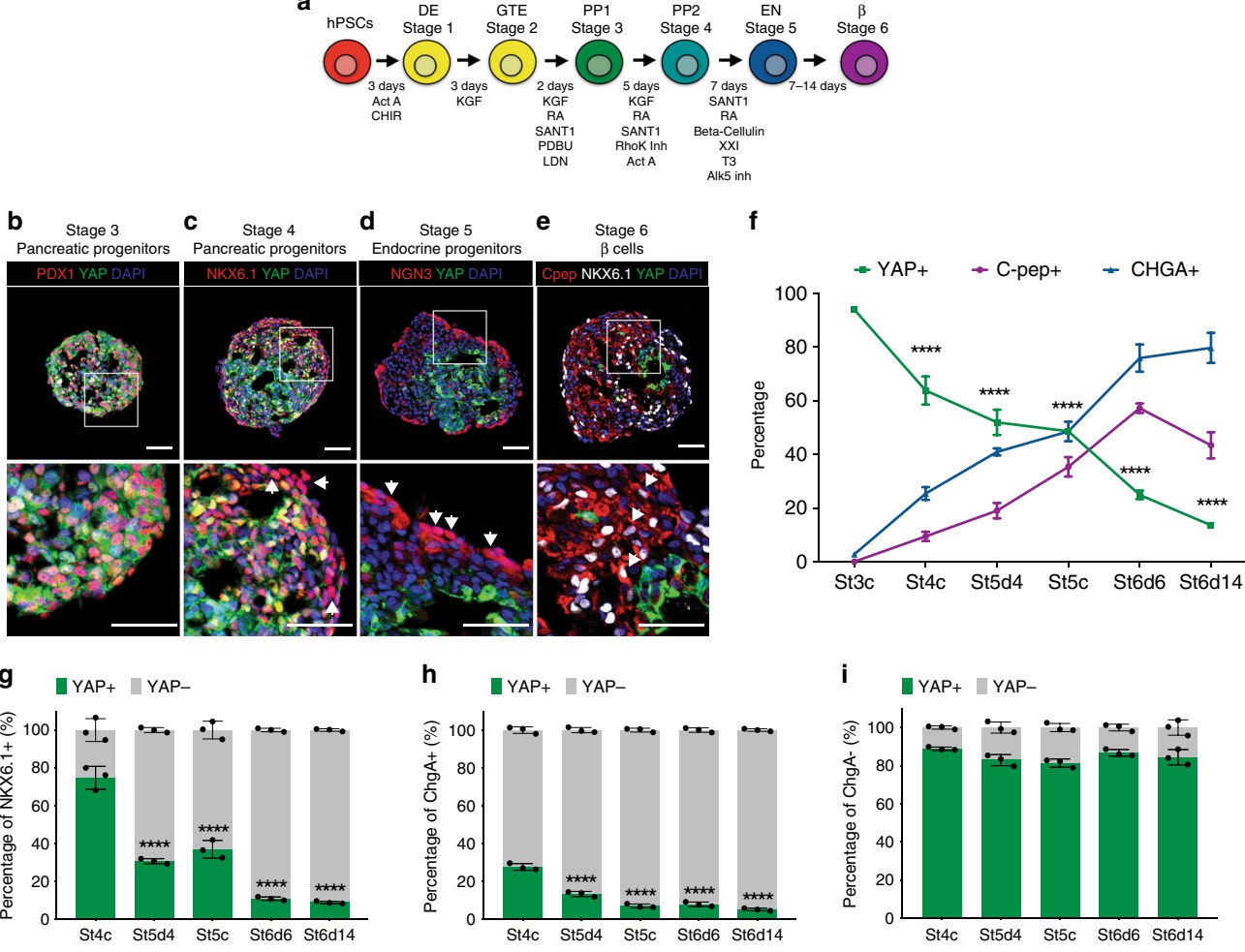

**Fig. 1** YAP downregulation in SC-endocrine and insulin-producing β cells. **a** Diagram of the directed differentiation of hPSCs into insulin-producing β cells. **b–e** Immunofluorescence micrographs of YAP expression in PDX1 + early pancreatic progenitors, NKX6.1 + late pancreatic progenitors, NGN3 + endocrine progenitors, and C-peptide + β cells. Representative images and cropped blots (bottom panels) are shown. Scale bar: 50 µm. White arrows denote NKX6.1 +/YAP1- (**c**), NGN3 +/YAP1- (**d**), and C-peptide +/NKX6.1 +/YAP1-cells (**e**). **f** Proportion of cells expressing YAP (green), C-peptide (purple) and CHGA (blue) from stage 3 through stage 6 as estimated by flow cytometry. **g–i** Proportion of YAP + and YAP- cells of all NKX6.1 + (**g**), CHGA + (**h**) and CHGA- (**i**) cells as quantified by flow cytometry. hPSCs human pluripotent stem cells, DE definitive endoderm, GTE gut tube endoderm, PP1 pancreatic progenitor 1, PP2 pancreatic progenitor 2, EN endocrine precursor, β insulin-producing β cells. Data represent mean values ± SEM, ****$p < 0.0001$, two-sided student's $t$ test ($n = 3$ biologically independent samples per group)

endocrine specification at stage 5, MPPs differentiated into NGN3 + endocrine progenitors more efficiently than DMSO controls, as assayed at stage 5, day 4 (Fig. 3b–d and Supplementary Fig. 2a; 12.1 ± 2% verterporfin treatment vs. 6.4 ± 0.52% control). Treatment with roscovitine, a cell-cycle inhibitor, at this stage did not have the same effect (Supplementary Fig. 2a, b), suggesting that blocking proliferation per se does not promote differentiation. The enhanced differentiation of NGN3 + endocrine precursors with verteporfin produces an increase in CHGA +/NKX6-1 + endocrine progenitor cells by the end of stage 5, something not observed with roscovitine treatment (Supplementary Fig. 2c–f, assayed at stage 5, day 7).

We then tested the effect of YAP inhibition from stage 4 through stage 6 of the β-cell differentiation protocol (Fig. 3a). YAP inhibition by verteporfin during these stages of differentiation leads to a significant increase in the proportion of C-peptide +/NKX6.1 + β cells (Fig. 3e–g and Supplementary Fig. 3a–h, 38.6 ± 3.9% verteporfin treatment vs. 27.5 ± 4.2% control). A modest but significant increase in glucagon +/C-peptide-alpha cells, but not somatostatin + delta cells, is also detected upon

YAP inhibition (Supplementary Fig. 3f–j). The effect of verterpofin on SC-endocrine and β-cell differentiation is robust and independent of the genetic background of the hPSC cell line (Supplementary Fig. 4).

Expression of YAP-targeted shRNAs during endocrine differentiation at stages 5 and 6 produces an increase in NGN3 (NEUROG3), PAX6, and NEUROD1 expression, as well as an increase in insulin (INS) expression (Fig. 3h; assayed at the end of stage 5). Similarly, MPPs expressing YAP-targeted shRNAs differentiated more efficiently into C-peptide +/NKX6.1 + β cells than progenitors expressing control shRNAs (Fig. 3i–l; assayed at the end of stage 6 differentiation, stage 6, day 14). Thus, YAP inhibition promotes the differentiation of MPPs into the endocrine lineage.

**Sustained activation of YAP impairs β-cell differentiation.** We overexpressed a stabilized form of YAP (YAPS6A)[27] in differentiating MPPs during the last two stages of differentiation (Fig. 4a). Lentiviral overexpression of YAPS6A resulted in a

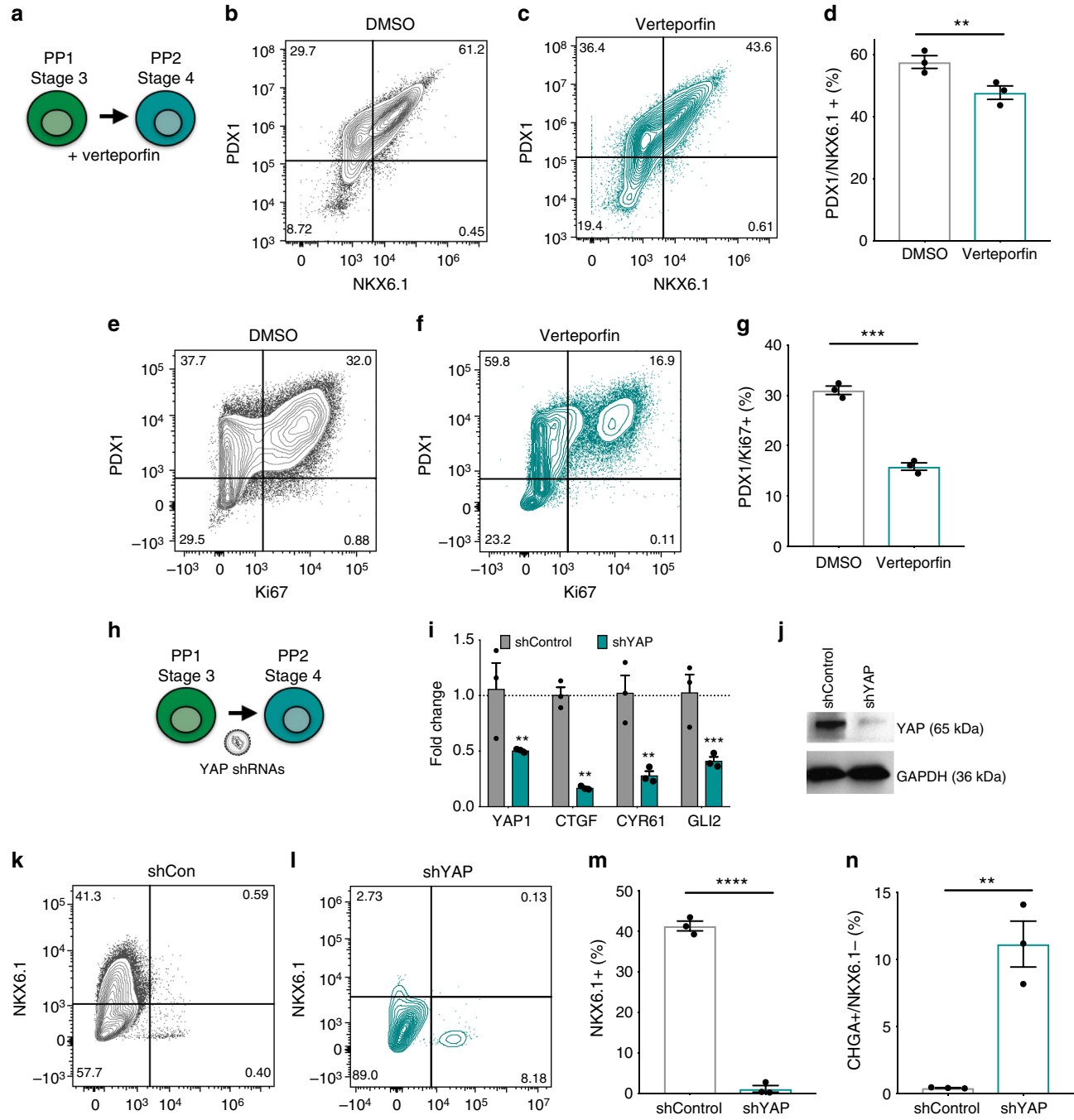

**Fig. 2** YAP activity regulates the specification and proliferation of NKX6.1 + progenitors. **a** Diagram of experimental design for **b**–**g**. **b**–**d** Flow cytometry of PDX1 and NKX6.1 expression in DMSO (**b**) or veterporfin-treated (**c**) MPPs and quantification of the proportion of PDX1/NKX6.1 co-positive pancreatic progenitors (**d**) from **b** and **c**, as assayed at the end of stage 4. **e**–**g** Flow cytometry analysis of PDX1 and the proliferation marker Ki67 in DMSO (**e**) and veterporfin-treated pancreatic progenitors (**f**), and quantification of coexpression of PDX1 and Ki67 in MPPs from **e** and **f** (**g**) as assayed at the end of stage 4. **h** Experimental design for **i**–**n**. **i**, **j** Effects of YAP shRNA expression during pancreatic differentiation on the expression of YAP and target genes assayed by qPCR for RNA (**i**) and western blot for protein (**j**), as assayed at the end of stage 4. qPCR values normalized to the average expression value of shControl samples. **k**–**n** Effects of the expression of non-targeting control (**k**) and YAP shRNAs (**l**) during pancreatic progenitor differentiation on NKX6.1 and CHGA expression and quantification of the proportion of NKX6.1 + (**m**) and NKX6.1-/CHGA + (**n**) cells from **i** and **j**, as assayed at the end of stage 4. Data represent mean ± SEM, \**p* < 0.01, \*\*\**p* < 0.001, \*\*\*\**p* < 0.0001, two-sided student's t test (*n* = 3 biologically independent samples per group)

sevenfold increase in the proportion of cells expressing YAP compared with LacZ controls (Fig. 4b, c; stage 6, day 14). Consistent with a role of YAP expression in limiting endocrine differentiation, YAPS6A overexpression leads to a significant decrease in the proportion of C-peptide + /NKX6.1 + β cells, compared with controls, with no effect on cell apoptosis

(Fig. 4d–f, m, n; stage 6, day 14). Verteporfin treatment rescued the impaired endocrine differentiation observed upon YAPS6A overexpression (Supplementary Fig. 6a–c). We quantified levels of proliferation of YAPS6A-overexpressing SC-β cells and detected a fivefold increase in the proportion of proliferating cells, compared with controls, as measured by coexpression of the proliferation

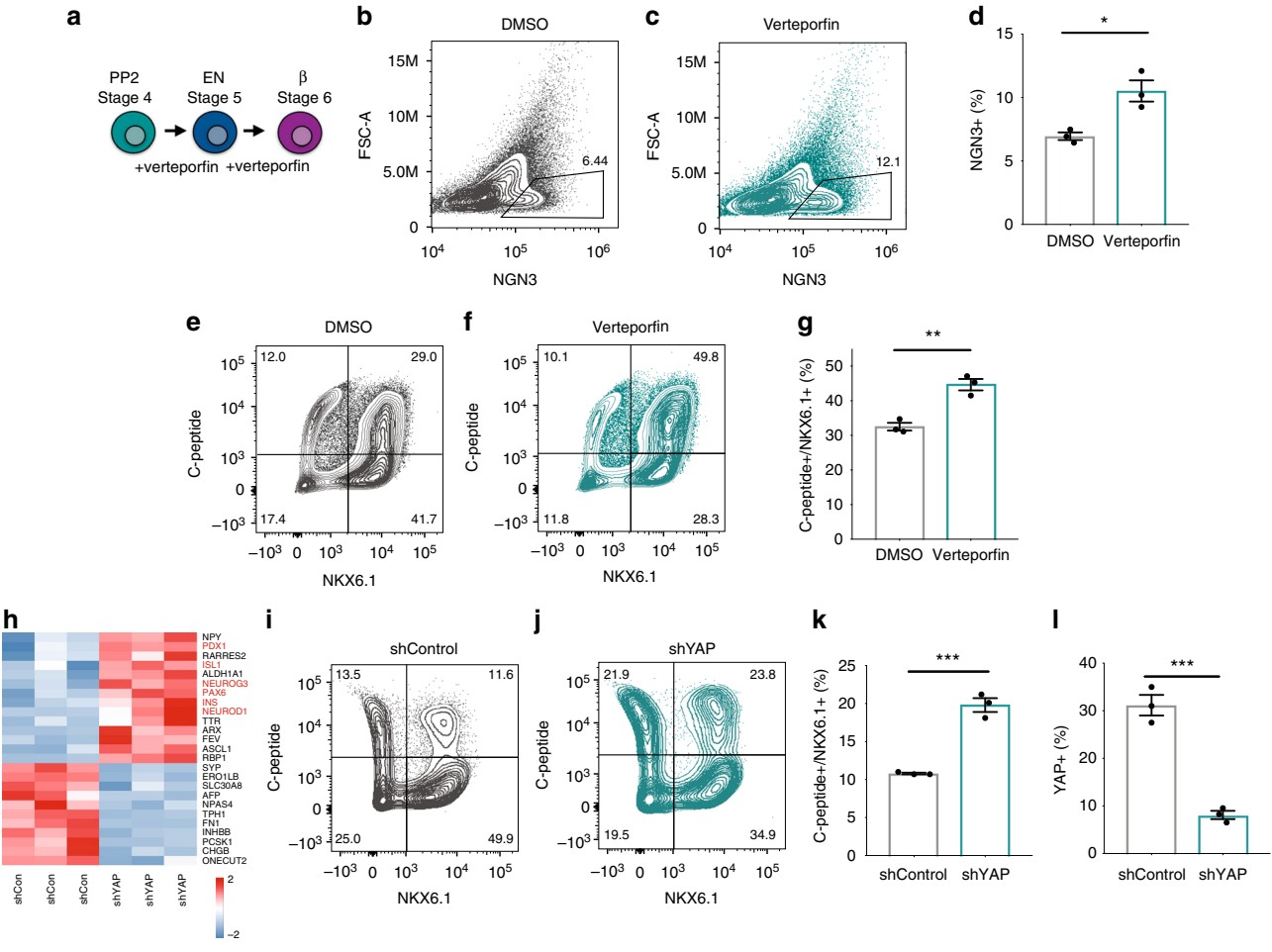

**Fig. 3** YAP inhibition enhances the generation of endocrine progenitors and SC-β cells. **a** Experimental design for **b–g**. **b–d** Flow cytometry analysis of NGN3 expression at stage 5, day 3 of DMSO and verteporfin-treated pancreatic progenitors and proportion of NGN3 + cells from **b** and **c**. **e–g** Flow cytometry analysis of NKX6.1 and C-peptide expression upon completion of the differentiation protocol of DMSO and verteporfin-treated pancreatic progenitors and quantification of the proportion of C-peptide + /NKX6.1 + SC-β cells from **e** and **f**. **h** Effects of non-targeting control and YAP shRNAs during endocrine differentiation on lineage marker mRNAs assayed by Nanostring assayed at the end of stage 5. Differentially expressed genes (adjusted p-value < 0.05) are displayed with genes relevant to endocrine induction highlighted in red. Data presented as z-scores. **i–k** Flow cytometry of C-peptide and NKX6.1 expression in non-targeting control and YAP shRNA-expressing cells and proportion of cells co-expressing both markers from i-j at the end of stage 6 differentiation (stage 6 day 14). **l** Proportion of cells expressing YAP in cultures of cells expressing non-targeting control and YAP shRNAs assayed at the end of stage 6 differentiation (stage 6 day 14) by flow cytometry. Data represent mean ± SEM, *p < 0.05, ***p < 0.001, two-sided student's t test (n = 3 biologically independent samples per group)

marker Ki67 with C-peptide and EdU incorporation in SC-β cells (Fig. 4g–l). This is consistent with previous reports[28,29]. The increase in proliferation was not specific to C-peptide + β cells as we observed an increase in the proliferation of non-β cells as well (Fig. 4j, l). These results demonstrate that YAP activity during endocrine differentiation limits the differentiation of MPPs into C-peptide + /NKX6.1 + β cells and, instead, promotes continued proliferation.

Functional β cells secrete insulin in response to glucose. Endocrine cells supplemented with either verteporfin or DMSO during stages 5 and 6 of differentiation (Fig. 3a) show an increase in insulin secretion at high glucose over low glucose in response to sequential glucose stimulations (Fig. 5a, b). Moreover, the levels of insulin secretion per total insulin content at 2.8 mM glucose, 20 mM glucose, and 30 mM KCl were significantly higher in cultures of SC-β cells differentiated with verteporfin (Fig. 5a and Supplementary Fig. 5). However, there were no statistically significant differences in stimulation indexes between control and verteporfin-treated endocrine cells (Fig. 5b). YAPS6A

overexpression during endocrine differentiation (Fig. 4a) hinders the ability of stem cell-derived β cells to secrete insulin in response to glucose (Fig. 5c, d). Importantly, YAPS6A-overexpressing β cells secreted insulin in response to KCl stimulation suggestive of defective GSIS in β cells. YAPS6A overexpression leads to a reduction in stimulation indexes compared with LacZ overexpression controls (Fig. 5d). Interestingly, a transient activation of doxycycline-inducible YAPS6A during the first 4 days of stage 6 leads to an increased proliferation of SC-β cells (Supplementary Fig. 6d–f), but does not affect the function of SC-β cells when assayed at stage 6, day 12 (Supplementary Fig. 6g, h). Thus, YAP inhibition may promote the differentiation of SC-β cells that have improved insulin secretion and sustained expression of YAP during endocrine differentiation restricts the differentiation of functional SC-β cells.

**Depletion of progenitor-like cells upon YAP inhibition.** In the adult mouse pancreas, YAP expression is restricted to

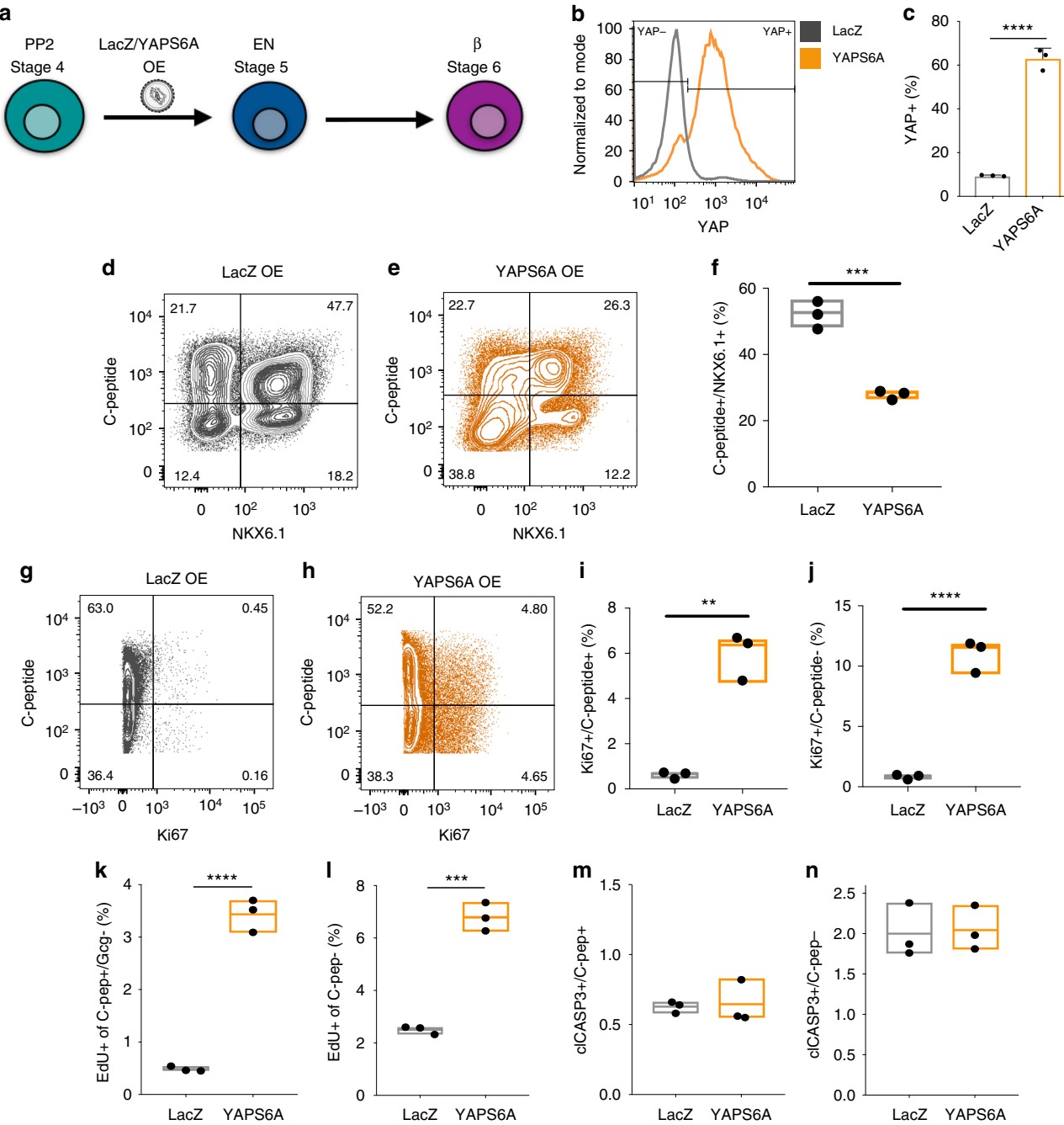

**Fig. 4** YAPS6A overexpression impairs differentiation of β cells and promotes proliferation. **a** Experimental workflow for the lentiviral overexpression of a stabilized form of YAP (YAPS6A) during β-cell differentiation. **b, c** Flow cytometry of YAP expression and quantification of YAP + cells in LacZ control and YAP-overexpressing stem cell-derived cells during β-cell differentiation (assayed at stage 6, day 14). **d–f** Flow cytometry analysis and quantification of C-peptide and NKX6.1 coexpression in LacZ and YAPS6A-overexpressing cells at the end of stage 6 differentiation (stage 6, day 14). **g–j** Flow cytometry analysis and quantification of C-peptide and Ki67 expression in LacZ and YAPS6A-overexpressing cells. **k–n** Quantification of EdU staining (**k, l**) and cleaved Caspase-3 staining (**m-n**) in monohormonal C-peptide + and C-peptide- cells by flow cytometry at the end of stage 6 differentiation (stage 6, day 14). Data represent mean (center line) ± min to max (bounds of box), **$p < 0.01$, ***$p < 0.001$, ****$p < 0.0001$, two-sided student's $t$ test ($n = 3$ biologically independent samples per group)

centroacinar and ductal cells and correlates with increases in levels of cell proliferation[23,30]. In all, 10–15% of the cells at the end of the β cell directed differentiation protocol express Ki67 and the ductal marker SOX9 (Fig. 6b, d, e, h). Given that a portion of SOX9 + ductal-like cells coexpress YAP (Supplementary Fig. 1h,i), we tested whether YAP inhibition would deplete this subpopulation of cells. Addition of verteporfin during the endocrine and β-cell differentiation stages (Fig. 6a) results in a decrease in the proportion of both SOX9 + ductal-like progenitor cells (Fig. 6b–d) and Ki67 + proliferating cells (Fig. 6e). Similarly, expression of YAP shRNAs during these stages leads to a three-fold decrease in the proportion of proliferative SOX9 + ductal progenitor cells (Fig. 6f–h).

Following transplantation, Sox9 + progenitor cells may significantly expand and form unwanted cells[19]. To test whether YAP inhibition depletes these proliferative cells in vivo, we

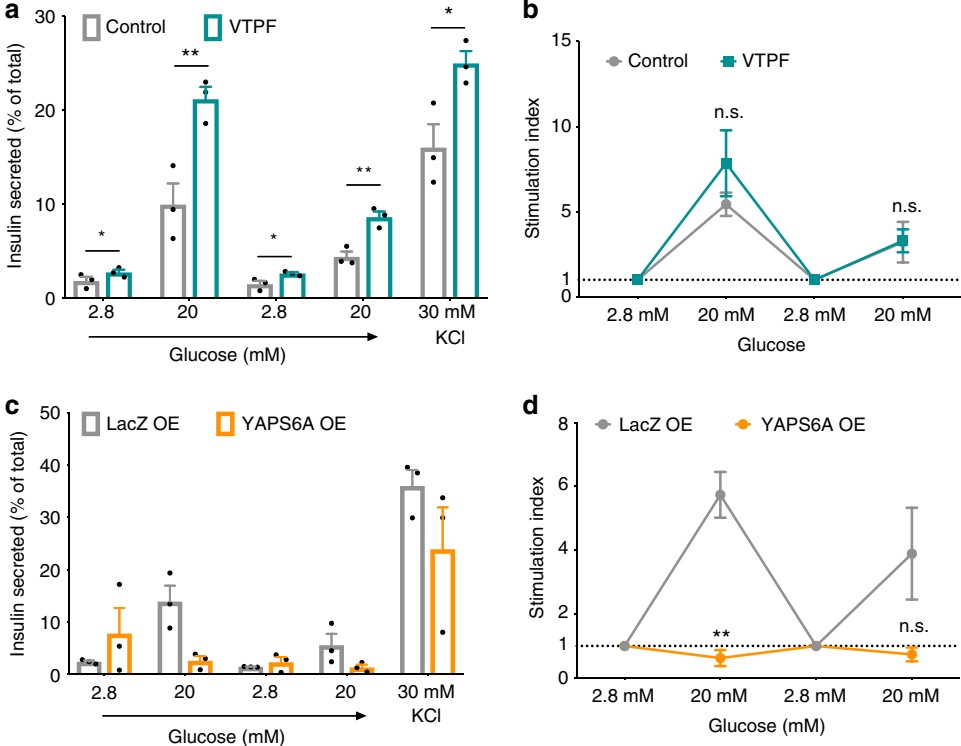

**Fig. 5** Functionality of SC-β cells upon YAP inhibition and overexpression. **a**, **b** Insulin secretion levels for DMSO or verteporfin-treated stem cell-derived β cells during sequential rounds of glucose and KCl challenge and insulin secretion stimulation indexes. Insulin secretion levels were normalized to total insulin content for each sample. Stimulation indexes were calculated as a ratio of insulin secretion at high glucose (20 mM) relative to the basal secretion (2.8 mM glucose). **c**, **d** Insulin secretion levels of LacZ and YAPS6A-overexpressing stem cell-derived β cells during sequential rounds of glucose and KCl stimulation and insulin secretion stimulation indexes. Data represent mean ± SEM, **$p < 0.01$, ***$p < 0.001$, ****$p < 0.0001$, n.s. non-significant, two-sided student's $t$ test, VTPF verterporfin ($n = 3$ biologically independent samples per group)

transplanted SC-β clusters into the kidney capsule of immuno-compromised mice[3]. Transplants of both control SC-β cells and cells differentiated in the presence of verteporfin (Fig. 6i) displayed robust in vivo glucose-stimulated human insulin secretion (Fig. 6j, k). At 12-week post transplantation, we quantified the proportion of SOX9 + cells in the kidney grafts and detected the expression of this marker in ~17% of the cells in control grafts (Fig. 6l–n). Transplants of SC-β cells differentiated in the presence of verterporfin displayed a reduction in the proportion of SOX9 + cells with an average of 3.1% in the grafts at this time point (Fig. 6m, n). The data suggest that YAP inhibition can partially deplete ductal-like and proliferative cells that may expand following transplantation.

## Discussion

The generation of functional SC-β cells from hPSCs relies on the modulation of signaling effectors, most of which were known to control in vivo pancreas organogenesis[3–5]. Efforts to elucidate novel cell fate determinants will not only shed light into this process but will also result in improved directed differentiation protocols for cell replacement therapies. Here we identify YAP, one of the effectors of the Hippo signaling pathway, as a factor involved in progenitor specification and differentiation into functional pancreatic endocrine cells. The regulation of YAP follows a developmental logic whereby its activity promotes the expansion of MPPs and its downregulation corresponds to their differentiation into endocrine progenitors and SC-β cells.

The Hippo signaling pathway controls organ size by coordinating progenitor proliferation and differentiation[21,31–33]. Mice with a pancreas-specific deletion of the upstream regulators MST1/2 displayed extensive acinar-to-ductal metaplasia and acute pancreatitis[22,23]. Sustained activation of YAP during pancreas development, or postnatally, resulted in a disrupted islet architecture and an undeveloped endocrine compartment[22,23,25]. However, dissecting lineage-specific dependencies on YAP activity during in vivo pancreas development is confounded by the extensive organ-wide tissue disarray[22,23]. Exploiting the in vitro differentiation of MPPs, the present study uncovers a role for YAP in limiting the differentiation of human endocrine cells by maintaining pancreatic progenitor identity and self-renewal. In agreement with this study, Cebola et al. showed that YAP and TEAD coactivators are core components of cis-regulatory modules of transcription factors that establish the multipotency and expansion of pancreatic progenitors, including SOX9, NKX6.1, GATA4, GATA6, HHEX, and FOXA2, among others[24]. Consistent with our data, other studies have shown that genetic downregulation or pharmacological inhibition of YAP impairs pancreatic progenitor proliferation[24,34].

MPPs undergo a cell-cycle lengthening and arrest during endocrine induction[8,11–13]. Cell cycle-dependent regulation of NGN3, an indispensable factor for pancreatic endocrine differentiation[7–10], further drives the specification of progenitors into the endocrine lineage[11,12]. However, the molecular underpinnings controlling the cell cycle-dependent specification into pancreatic endocrine lineages remain elusive. The present study lends support for a salient role for YAP in establishing whether a progenitor cell undergoes proliferation or differentiates, similar to the cell cycle-dependent differentiation models put forward for neural differentiation[35,36]. We also show that a downregulated activity of YAP, either chemically or genetically, enhances the generation of endocrine cells and SC-β cells. Part of this process

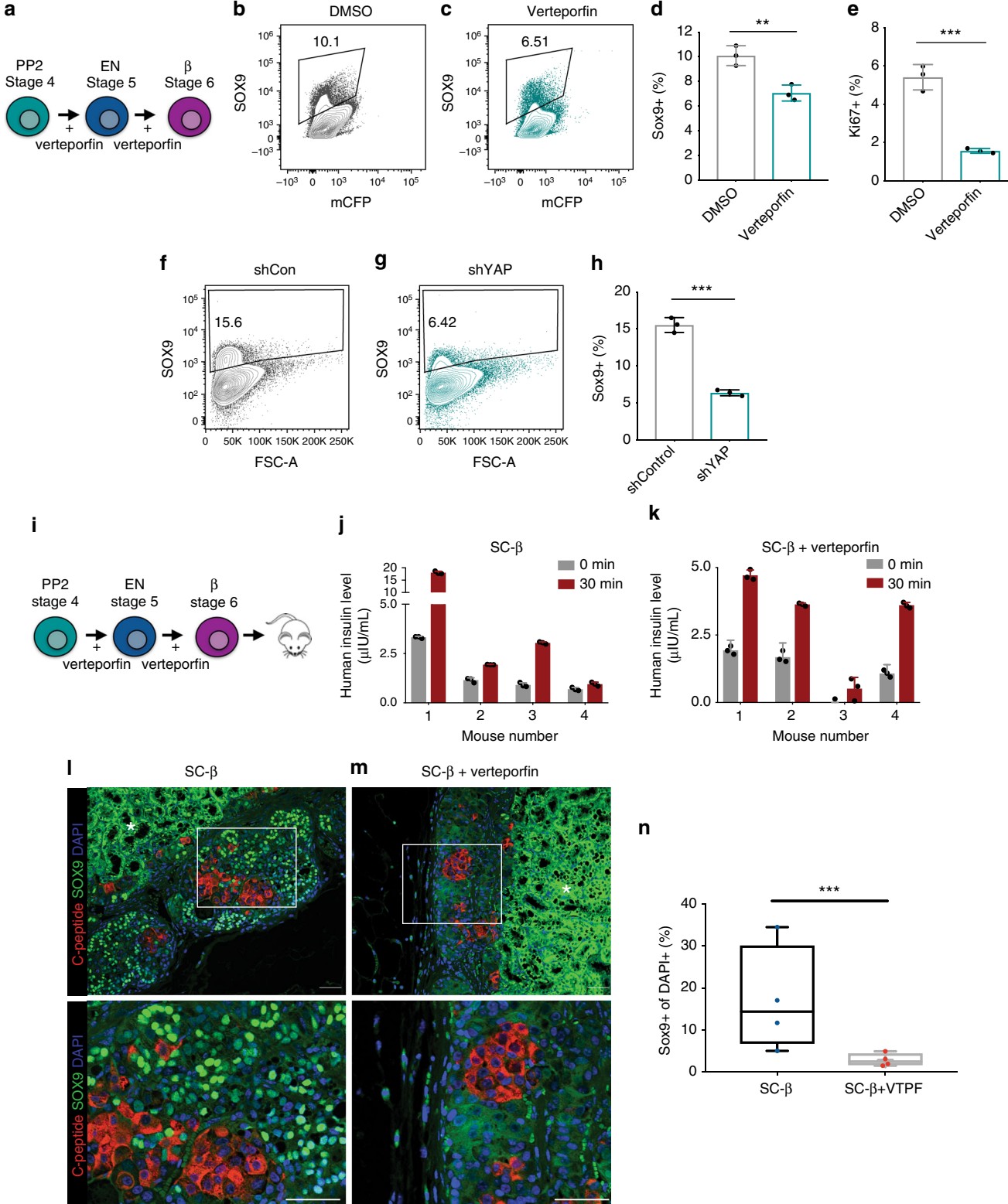

may be the downregulation of SOX9 upon YAP inhibition. SOX9 is essential for the maintenance of mitotically active MPPs and ductal differentiation during pancreas organogenesis[37,38] and its expression is controlled by YAP in other cell types[32,39]. As most YAP + cells at the end of our directed differentiation protocol coexpress SOX9, we further posit that a failure to induce YAP downregulation in MPPs may limit their differentiation into endocrine lineages and contribute to the cellular heterogeneity

observed during in vitro differentiation of SC-β cells[3,4]. Further support for this interpretation comes from a recent study that uncovered a mechanotransduction-dependent role of YAP in ductal and endocrine lineage bifurcation[25].

In all, we propose that a dual modulation of YAP activity can improve the directed differentiation into the β-cell lineage. We have provided evidence for the utility of the YAP inhibitor verteporfin as a potent inducer of endocrine differentiation and

**Fig. 6** YAP inhibition reduces Sox9 + progenitors in vitro. **a** Experimental design for **b–e**. **b–d** Flow cytometry of SOX9 expression in the control and verteporfin-treated differentiating progenitors and quantification of the proportion of SOX9 + cells from **b** and **c** at the end of stage 6 differentiation (stage 6, day 14). mCFP is an unstained control. Data represent mean ± SEM, **p < 0.01 (n = 3 biologically independent samples per group). **e** Proportion of Ki67 + in control or verteporfin-treated differentiating progenitors as assayed by flow cytometry at the end of stage 6 differentiation. Data represent mean ± SEM, ***p < 0.001 (n = 3 biologically independent samples per group). **f–h** Flow cytometry analysis of SOX9 in non-targeting control and YAP shRNA-expressing pancreatic progenitors and quantification of SOX9 + cell proportion (**h**) at the end of stage 6 differentiation. Data represent mean ± SEM, ***p < 0.001 (n = 3 biologically independent samples per group). **i** Experimental design for **j–n**. **j, k** in vivo GSIS of control and verteporfin-derived SC-β cell grafts as assayed by serum human insulin before and 30 min after a glucose injection 8 weeks post transplantation. **l, m** Immunofluorescence micrographs of grafts 12 weeks post transplantation of control and verteporfin-derived SC-β cell grafts stained for C-peptide and SOX9. Representative images and cropped blots (bottom panels) are shown. Asterisk (*) denotes nonspecific staining. Scale bar: 50 μm. **n** Quantification of SOX9 + of DAPI + cells within grafts 12 weeks after transplantation. Data represent mean (center line) ± min to max (whiskers) and lower and upper quartiles (bounds of box), ***p < 0.001, two-sided student's t test (n = 4 animals per group, values correspond to the average proportion of SOX9 + cells from four histological sections per animal)

progenitor depletion in vitro. In addition, we hypothesize molecules that enhance YAP activity during the specification of MPPs may promote the expression of NKX6.1 in this population of progenitors, as our data link YAP activity to its expression, critical for the formation of monohormonal β cells at later stages of differentiation[3,4,14,15]. Given the known role of Hippo signaling and YAP during organ development[20], this approach may be applicable to other directed differentiation models that guide the differentiation of hPSCs into post-mitotic cells.

## Methods

**Cell culture and differentiation of hPSCs.** hPSCs were maintained in mTeSR1 (Stem Cell Technologies) in 500 -mL spinner flaks on a stir plate (Chemglass) set to 70 rpm in a 37 °C incubator, 5% $CO_2$, and 100% humidity. All the experiments were carried out using the human embryonic stem cell line HUES8 and the induced pluripotent stem cell lines (iPSC) 1016 and 13B[3,40]. Cell lines were obtained from the Human Embryonic Stem Cell Facility and iPS Core Facility of the Harvard Stem Cell Institute, and University of Massachusetts Medical School.

Differentiations into SC-β cells were performed following a protocol described by our group[3] as follows: HUES8 or 1016 cells were seeded at $6 \times 10^5$ cells/mL in mTeSR1 media and 10 μm Y27632 (Sigma-Aldrich). The media was changed 48 h later and the differentiations were started 72 h after the cells were seeded. The media changes were as follows:

Stage 1 DE: Day 1: S1 + 100 ng/mL ActivinA (R&D Systems) + 3 μM Chir99021 (Stemgent). Day 2: S1 + 100 ng/mL ActivinA.

Stage 2 GTE: Days 4 and 6: S2 + 50 ng/mL KGF (Peprotech).

Stage 3 PP1: Days 7 and 8: S3 + 50 ng/mL KGF + 0.25 μM Sant1 (Sigma) + 2 μM RA (Sigma) + 200 nM LDN193189 (only Day 7) (Sigma) + 500 nM PdBU (EMD Millipore).

Stage 4 PP2: Days 9, 11, and 13: S3 + 50 ng/mL KGF + 0.25 μM Sant1 + 100 nM RA + 10 μm Y27632 + 5 ng/mL ActivinA.

Stage 5 EN: Days 14 and 16: S5 + 0.25 μM Sant1 + 100 nM RA + 1 μM XXI (EMD Millipore) + 10 μM Alk5i II (Axxora) + 1 μM T3 (EMD Millipore) + 20 ng/mL Betacellulin (Thermo Fisher Scientific). Days 18, 20: S5 + 25 nM RA + 1 μM XXI + 10 μM Alk5i II + 1 μM T3 + 20 ng/mL Betacellulin.

Stage 6 β: Days 21–35 (change every other day): S3 media. In the final stage, cells were analyzed between 28 and 35 days of the protocol.

Control and experimental samples were collected from differentiations performed in parallel with hPSC cells obtained from the same maintenance batch. Treatments with 0.35 μM verteporfin (Sigma-Aldrich), 10 μM roscovitine (Sellekchem), or DMSO (Sigma-Aldrich) were performed at the last stages of the differentiation including stage 4 (days 9–14), stage 5 (days 14–18), and stage 6 (day 18–25, first 7 days) in ultra-low attachment six-well plates (Corning).

**Lentivirus production and infection of cells.** Lentiviral particles were produced by transfecting 293 T cells (Takara Bio) with the packaging vectors pHDM-vsvg, pHDM-tat, pHDM-rev, and pHDM-gag/pol along with lentiviral backbone vectors using the TransIT-293 transfection reagent (Mirus). The lentiviral vectors used in this study were the following: YAP1(S6A)-pLX304 (addgene #42562), LacZ-pLX304 (addgene #42560), YAPS6A-pLIX403, GFP-pLIX403, 29-mer scrambled shRNA in pGFP-C-shLenti (Origene), and four unique 29mer YAP shRNAs in pGFP-C-shLenti (Origene). Lentiviral particles were concentrated 48 h and 72 h post transfection using the PEG-IT virus precipitation reagent (Fisher Scientific) overnight at 4 °C followed by centrifugation at 1500 g for 30 min at 4 °C and stored at −80 °C.

For infection, cell clusters collected from spinner flask suspension cultures were dissociated in Accutase (StemCell Technologies) for 10 min, followed by mechanical dissociation and centrifugation at 230 g for 5 min at room temperature

(RT). Cell pellets were resuspended at a density of 2.5 million cells/mL in the stage-matched medium with polybrene reagent (Santa Cruz) at 8 μg/mL. Single-cell suspensions were combined with concentrated lentiviral particles and plated on ultra-low attachment six-well plates on a rocker plate set at 70 rpm in a 37 °C incubator, 5% $CO_2$, and 100% humidity. Feeding schedule was followed depending on the stage of differentiation as described above. For the transient induction of GFP and YAPS6A expression, infected cells were treated with 2 μg/mL doxycycline (Sigma-Aldrich), as described here.

**Immunohistochemistry.** Cell clusters were collected at the end of each of the differentiation stages and fixed with 4% paraformaldehyde for 1 h at RT, washed in 30% sucrose, embedded in OCT compound (Tissue-Tek) and sectioned for histological analysis. Clusters were blocked with PBS + 0.1% Triton X-100 (VWR) + 5% donkey serum (Jackson Immunoresearch) for 30 min at RT, incubated with primary antibodies overnight at 4 °C, washed, and incubated with secondary antibodies for 1 h at RT. Stained clusters were washed and mounted in Flouromount-G (Invitrogen), covered with coverslips, and sealed with nail polish. Images were taken using a Zeis Axio Imager.Z2 and Apotome.2. Image analysis was performed with ImageJ. To estimate the relative proportion of cells in the graft, four tissue sections from different areas in the graft were imaged, and the relative proportions of relevant cell types were quantified as a percent of cells positive for the relevant marker of the total number DAPI nuclei. The antibodies used in the study were the following: rabbit anti-YAP (1:100, Cell Signaling Technology; 14074 S), mouse anti-YAP (1:100, Abnova; 89106308), rabbit anti-SOX9 (1:100, Cell Marque; AC-0284RUO), rat anti-C-peptide (1:200, Developmental Studies Hybridoma Bank; GN-ID4), mouse anti-NKX6.1 (1:100, Developmental Studies Hybridoma Bank; F55A12-supernatant), rabbit anti-Ki67 (1:100, Abcam; ab16667), sheep anti-NGN3 (1:50, R&D systems; AF3444), goat anti-PDX1 (1:100, R&D systems; AF2419), rabbit anti-CHGA (1:200, Novus Biologicals; NB120–15160), mouse anti-CHGA (1:200, Santa Cruz; sc-393941), mouse anti-glucagon (1:200, Abcam; ab82270), mouse anti-somatostatin (1:200, Santa Cruz; sc-55565), rabbit anti-cleaved Caspase-3 (1:100, Cell Signaling; 9661) and mouse anti-PCNA (1:100, Millipore; NA03).

**Flow cytometry.** Differentiated cell clusters were dispersed into a single-cell suspension with TrypLE (Life Technologies) at RT, fixed with 4% paraformaldehyde at 4 °C, washed three times in PBS + 0.2% bovine serum albumin (Millipore) + 0.1% saponin (Sigma), blocked for 30 min at 4 °C in PBS + 5% donkey serum + 0.1% saponin, incubated with primary antibodies overnight at 4 °C, washed, and incubated with secondary antibodies for 1 h at RT. Stained, fixed cells were filtered through a 40- μm nylon mesh into flow cytometry tubes (BD Falcon) and were analyzed for relevant stage-specific marker expression using LSR II flow cytometers (BD Biosciences) and FlowJo for the data analysis. Samples stained with secondary antibodies only were used to gate on cells that are negative and accurately identified cells expressing the markers included in the analysis. For shRNA-infected samples, only infected GFP + cells were included in the analysis (approximately 30–40% of all cells). To quantify incorporation of EdU and estimate levels of proliferation, cells pulsed with 10 μM EdU for 4 h were dissociated as described above unless stated otherwise. The Click-IT EdU flow cytometry kit (Invitrogen) was used to detect EdU incorporation following the manufacturer's protocol with minor modifications.

**Glucose-stimulated insulin secretion.** Approximately $1 \times 10^6$ SC-β cells in clusters were washed with Krebs buffer and incubated in low (2.8 mM) glucose Krebs in cell culture inserts (Millicell) to starve cells and remove residual insulin at 37 °C for 1 h. Clusters were washed and incubated in low glucose Krebs for 1 h and the supernatant was collected. They were then transferred to high (20 mM) glucose, incubated for 1 h and the supernatant was collected. For sequential GSIS challenges, this sequence of low- and high-glucose stimulations was repeated twice. At the end, clusters were incubated in 2.8 mM glucose + 30 mM KCl Krebs

(depolarization challenge) for 1 h and the supernatant was collected. Clusters were dispersed into single cells using TrypLE and cell number was estimated by a Vi-Cell counter (Beckman Coulter). Insulin concentration was determined for supernatant samples using the Human Ultrasensitive Insulin ELISA (ALPCO Diagnostics). Protein extraction was performed with the M-PER extraction reagent (Thermo Scientific), and insulin content was measured for each sample using the human Ultrasensitive Insulin ELISA kit. Insulin secretion levels were normalized to total insulin content for each sample. Stimulation indexes were calculated as a ratio of insulin secretion at high glucose (20 mM) relative to the basal secretion (2.8 mM glucose).

**Mouse transplantation analysis**. Dissociated SC-β cells were transplanted into the kidney capsule of immunodeficient SCID-beige mice (Jackson Laboratory), aged 8–10 weeks[3]. In total, $5 \times 10^6$ differentiated cells (per animal) collected at the end of the SC-β cell differentiation protocol were dispersed using Accutase and resuspended in 200 μL of the RPMI1640 medium and kept on ice for 5–10 min before loading into a catheter for cell delivery under the kidney capsule. Mice were first anesthetized with 0.5 mL/25 g 1.25% avertin/body weight, and the left ventral site was shaved and betadine and alcohol was applied to clean the incision site. A 1 -cm incision was performed to expose the kidney for the insertion of the catheter needle and injection of the cells. The abdominal cavity was closed with PDS absorbable sutures (POLY-DOX) and the skin was closed with surgical clips (Kent Scientific Corp). Mice were placed on a 37 °C micro-temp circulating pump and blanket during surgery and recovery period and given a pre-emptive dose of 0.2 mg/kg Meloxicam along with a dose of 0.1 mg/kg Buprenorphine immediately after surgery, and one additional dose of Meloxicam 24 h later. Wound clips were removed 10 days after surgery and mice were monitored twice a week.

Mice were then analyzed for graft function at various time points by performing in vivo[3]. After fasting the mice for 16 h overnight, a glucose challenge was performed by intraperitoneal injection (IP) 2 g D-( + )-glucose/1 kg body weight and blood was collected both pre-injection and 30 min post-injection of glucose through facial vein puncture. Serum was separated out using Microvettes (Sarstedt) to then measure human insulin levels using the Human Ultrasensitive ELISA kit. Kidney grafts were dissected from the mice, fixed in 4% paraformaldehyde overnight, embedded in paraffin, and sectioned for the histological analysis as described above.

All animal experiments were performed in accordance with Harvard University International Animal Care and Use Committee (IACUC) regulations.

**RNA analysis**. The total RNA was isolated using the Direct-ZOL Miniprep kit (Zymo Research), and RNA was stored at −80 °C in RNA storage solution (Invitrogen) until subsequent analysis. For real-time PCR, cDNA was synthesized using the SuperScript II Reverse Transcriptase kit (Invitrogen) and anchored oligo-dT primers (Invitrogen). TaqMan Fast Universal-based real-time PCR (Life Technologies) was performed in a ABI 7900HT PCR system (Applied Biosystems), with commercially available TaqMan gene expression assay probes (Thermo Scientific) with GAPDH as an internal normalization control (Supplementary Table 1). For NanoString analysis, 50–100 ng of RNA was hybridized to a custom nCounter XT probe set and processed using the NanoString prep station and nCounter (NanoString Technologies). Gene expression levels were determined with NanoString nSolver software with default parameters and normalized with the expression of five housekeeping genes (ITCH, RPL15, RPL19, TCEB1, and UBE2D3). Nanostring data have been deposited into figshare under DOI accession code 10.6084/m9.figshare.7670531.

**Western blot analysis**. Dispersed cells were lysed in RIPA buffer (Thermo Scientific), and protein concentration was measured using the BCA Protein Assay kit (Thermo Scientific). In total, 5–10 μg of protein extracts were separated by AnyKD Mini-Protein precast gels (Bio-Rad) and transferred to nitrocellulose membranes (Bio-Rad). Membranes were blocked in 3% BSA + 0.1% Tween 20 TBS for 30 min at RT and then incubated with the following primary antibodies overnight at 4 °C: rabbit anti-YAP (Cell Signaling; 14074S) and mouse anti-GAPDH (Millipore; MAB374) as the loading control. After washing, the membranes were incubated with HRP-conjugated secondary antibodies for 1 h at RT, and then incubated in chemiluminescent ECL detection reagent (VWR) for signal detection and development. Uncropped scans of western blots presented in Fig. 2 are provided in Supplementary Fig. 7.

**Statistical analysis**. Statistical analysis was performed using unpaired two-sided $t$ tests, unless stated otherwise. For all the experiments included in this study, three or more biological replicates were performed using stage-matched controls as a reference. The Bonferroni–Dunn method was used for adjustment of multiple comparison analysis of qPCR and Nanostring data using GraphPad PRISM 7. Box plots, bar graphs, and heatmaps were generated with GraphPad PRISM 7 and R.

**Reporting Summary**. Further information on experimental design is available in the Nature Research Reporting Summary linked to this article.

## Data availability

The datasets generated during and/or analyzed during the current study are available from the corresponding author on reasonable request.

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

## Acknowledgements

We thank Ronny Helman, Jose Rivera-Feliciano, Stephanie Tsai, Aubrey Faust, Nadav Sharon, Juan Alvarez, and Adrian Veres for helpful discussions. We also thank members of the Harvard Stem Cell Institute Flow Cytometry and Histology core for technical support. E.R.O. was supported by a fellowship from the National Science Foundation. D.A.M. is an investigator of the Howard Hughes Medical Institute. This work was supported by grants from the Harvard Stem Cell Institute, NIH, and the JPB foundation.

## Author contributions

E.R.O., K.A., and D.A.M. designed, carried out, and analyzed experiments. J.K. supervised in vivo transplantation studies. E.R.O. and D.A.M. wrote the paper with contributions from K.A. All authors reviewed, edited, and approved the paper.

## Additional information

**Competing interests:** A patent application has been filed on aspects of this work by E.R. O. and D.A.M. The remaining authors declare no competing interests.

