## [Peer Review File · Nature Communications]

Reviewers' comments:

Reviewer #1 (Remarks to the Author):

In this manuscript, Rosado-Olivieri et al show that YAP, a downstream effector of the Hippo signalling pathway, restrains pancreatic endocrine progenitor differentiation, and that inhibition can be exploited to enhance in vitro generation of beta cells. It also confirms earlier work showing that YAP promotes the expansion of pancreatic multipotent progenitors. Interestingly, it provides evidence to support that the effect of YAP is specific, inasmuch as it is not mimicked by other maneuvers to block proliferation. This is an important study that provides proof-of-principle evidence to implicate a new pathway in efforts to differentiate beta cells from iPSC. Some aspects should be addressed for publication:

Major points:

- Fig. 1. Many YAP+ cells appear to show cytoplasmic, non-nuclear patterns. This is not indicative of active YAP, and should be assessed in a more quantitative manner.
- It is difficult to assess the downregulation between stages 4 (NKX6.1 late PP) and stage 5 (endocrine progenitors) based on representative immunofluorescence images without more quantitative methods such as flow cytometry.
- Fig. 2. Is the increase in Chga+ cells observed with both shRNAs and chemical inhibition? What is the nature of these CHGA+ NKX6-1+ cells, are they glucagon + cells? Unless this is clarified it might make sense to either add a qualifying statement or remove these findings from this Figure.
- Figure 2, and other figures: Y axes should start at zero to avoid giving the impression that effects are larger than they really are.
- The authors state that YAP inhibition with verteporfin leads to a significant decrease in the number of NKX6.1 progenitors. Can the authors confirm that the 28% reduction is the average reduction of the 3 replicates used and provide a measure of variation and statistical significance? Same question for Figure 3g and 40,4% increase in C-peptide/NKX6.1+ cells. Numbers in the text do not seem to match Figure 3g. Same concern for 17% reduction in Sox9+ cell reduction in Figure 6
- Fig. 4h. This seems to show number of C-peptide + cells/Ki 67+ (with Ki67 as the denominator). Since the number of C peptide+ cells increases it seems more appropriate to show Ki67+/C-peptide+ cells.
- Figure 6. It is unclear if authors normalized the number of INS+ transplanted. Either way this should be clarified. If the increase in differentiation of endocrine cells is offset by a decrease in SOX9+ progenitors, the next question is what is the effect of verteporfin on the hypothetical pool of cells that are neither bona-fide multipotent progenitors or endocrine cells?
- The discussion should perhaps clarify the authors view on whether

Minor comments

Figure 4b Minor figure editing is needed: Y-axis labels should be enlarged. Currently unreadable Stars representing statistical significance in Figure 5a and 6n as well as individual data points from Figure 5c should be enlarged.

Calculation of stimulation index from GSIS assays is not described in Methods

- .

Reviewer #2 (Remarks to the Author):

The data in this paper support an important but incremental advance in our knowledge of beta cell differentiation and development. YAP inhibition promotes increased generation of pancreatic progenitors, with some possible impact on beta cell function. See below for specific comments and concerns.

Concerns

1- Figure 1: I am not sure that IF is the best way to characterize YAP expression in the different stages and populations of beta cell differentiation. Is this image representative of what is going on more broadly? I would like to see either FACS analysis or mRNA expression in sorted cell populations in addition to IF: this is shown only for stage 5 cells. A more comprehensive time course is necessary, especially as this is the basis for the hypothesis driving the rest of the paper. All FACS data should be aggregated from multiple experiments and analyzed with the appropriate statistical test to demonstrate any difference is real.

Figure 1G is certainly unconvincing as a FACS plot of YAP positive versus YAP negative cells; more controls justifying this gating are needed.

1- Figure 2J: the loading controls are extremely different, making evaluation of YAP protein knockdown problematic. The bands for the YAP protein blot do not look side-by-side because the backgrounds are different, whereas the control bands do appear side-by-side.

2- Figure 3 A-D. Flow cytometry for NGN3 suggest an increase of NGN3+ population. Can this finding be confirmed by RNA or western blot.

-Figures 3E-G: the quantification in G is more impressive looking than the FACS plots in E and F would suggest. I would like to see more FACS plots in supplemental figures with appropriate controls to visually support the quantification.

- Figure 3E-G and 3I-K: why is the yield of C-peptide and Nkx6.1 double positive cells so different in controls between experiments?

- Use of cell cycle inhibitor roscovitine (Figure S2) was a good parallel experiment, but I would like to see more replicates as there did appear to be a trend towards a difference: an n of 3 may be underpowered.

-In addition the absolute yield of c-peptide+NKx6.1+ cells increased due to YAP inhibition should be shown, not just percentage.

3- Figure 4F-G: Ki67 expression seems to vary along a continuum rather than be a bimodal process, which is interesting and not addressed. Has this previously been established? I would like to see more FACS controls in supplemental figures to support and follow-up on this result. Cell yields and markers of apoptosis would be helpful to see what exactly YAP is doing. Ki67 staining does not necessarily equal proliferation.

-In addition, an inducible overexpression of the stabilized YAP would be very helpful, as it could

demonstrate if 1- can this method be used to meaningfully expand ES derived beta cells or progenitors, and 2- do these expanded cells generate functional beta cells once the YAP overexpression is stopped. Possibly adding the inhibitor to their OE cells may be a method to do this if the drug is still functional on the stabilized YAP.

4- Y axes on Figure 5A and 5C are 10-fold different, raising concerns about normalization. I would like to see the raw numbers (level of insulin secretion and total insulin content) graphed independently in supplemental figures to support the conclusions. Insulin/c-peptide secreted per insulin+ cell is another useful way to show these sorts of data. It looks that the inhibitor treated cells just secrete more insulin in all circumstances but SI does not seem to be impacted? There is no indication of a statistical test being performed in Figure 5b.

5- While GCG was examined in the supplemental materials, what about SST expression upon modulation of YAP pathway during differentiation? Considering that inhibitor treated cultures secrete more insulin in general, loss of SST could be a possible mechanism for this finding.

6- The authors state they use two different genetic backgrounds but they only showed data from one cell line and it was not clear if the results are similar in both genetic backgrounds. Data from the second genetic background must be presented in supplemental materials for all major findings both in differentiation and function, with statistical tests performed to demonstrate that these effects are reproducible. Honestly, a third line would be very helpful for the most critical experiments such as drug inhibition on differentiation efficiency and beta cell function. The beta cell differentiation field has had issues with protocol reproducibility and it is critical to demonstrate these effects work in multiple genetic backgrounds.

minor issues

1- I dislike the frequent (but not ubiquitous) use of non-0 y axis origins to make data look more significantly different, though I suppose passing the statistical test is more important than any graph.

2- Overall, it is not always clear from the legends what stage/day cells are being analyzed and there seem to be internal inconsistencies in yields or cell numbers from experiment to experiment.

3- Though verteporfin is an established inhibitor of YAP, I would have liked to see at least one control (FACS plot or Western blot) showing this.

4- It is unclear if YAP's promotion of cell proliferation is a process specific to the beta cell or pancreatic progenitor or endoderm or is even a general process, especially given the conclusions of figure 6. This may not be able to be addressed experimentally (or may be beyond the scope of this paper), but warrants further discussion.

5- In figure 2 H-N. shRNA experiments do not show virus infection efficiency. It is really hard to conclude that shRNA-YAP drives to strong phenotype in NKX6.1+ population and proliferation without knowing what percentage of cells are carrying the virus.

6- Y axis labelling is confusing in figure 5b. Why is the dotted line (presumed baseline) on 1.5 SI. This is generally shown at 1.

Reviewer #3 (Remarks to the Author):

Review of YAP inhibition enhances the differentiation of functional stem cell-derived insulin-producing β cells by Rosado-Oliveri et al.

This manuscript explores the proposition that the TAZ/YAP pathway plays a role in pancreatic differentiation in vitro from pluripotent stem cells. During mouse pancreatic development, other studies have suggested that this is so, and therefore, there is some currency in the idea that manipulation of this pathway in vitro, in the right circumstances, could enhance the yield or quality of beta cells so derived.

Figure 1 Outlines the general differentiation protocol and performs a basic characterisation of this story's key protagonists over the course of differentiation. Attention is focussed on NGN3, YAP and NKX6.1. As will be unveiled in later figures, the relationship between YAP and NGN3 expression is of particular interest, with the potential observation that YAP expression is enriched in NGN3+ cells. This relationship is key to much of what follows, elevating the importance of this figure. The strength of the idea is based on immunofluorescence staining and intracellular flow cytometry, the latter of which can be a technique fraught with traps - particularly with poorly characterised antibodies or antiserum. Close examination of the image of cells at the endocrine progenitor stage indicates that much of the NGN3 staining is cytoplasmic. As far as I'm aware, cytoplasmic NGN3 is not commonly observed - and so, I believe it is incumbent upon the authors to show some additional validation that this staining is real. Turning to the intracellular flow cytometry, it's not clear what rationale was used to place the gates in their positions. Some explanation of this is required. Given that the primary antibody in this experiment is in fact a polyclonal sera, it would be very useful, necessary even, for the authors include a non specific control (say a sera directed against something else - that isn't expressed) to be confident about why and where these gates are set. This same argument applies to the YAP staining. In addition, I think it is necessary to show, probably using Q-PCR, that the proteins assayed by FACS are really enriched or otherwise in the various fractions. Until this is done, it's very difficult to unconditionally accept the data as it's presented. Parenthetically, the NGN3 staining in panel "i" looks far more convincing - but also, it does not look like 20%. Last, as with figures to follow, it's very difficult to figure out exactly how many times these experiments have been done - particularly with respect to the flow cytometry experiments.

Figure 2 looks at the potential of YAP using a chemical inhibitor, verteporfin. Again, the position of FACS gates is something that needs to be addressed. At this point the number of experimental repeats also needs to be explicitly stated. The figure legend says n=3 per group - but in itself, this statement conveys insufficient information. This data should probably be presented as the mean and SEM for 3 independent experiments. If this is not the case then this needs to be stated. Similarly as for figure 1, there needs to be some independent confirmation that the FACS is reading out reality - something that could be accomplished using Q-PCR on sorted fractions. The other clear deficit with this figure is the absence of FACS showing the reduction YAP expression following introduction of shRNAs - this is so obvious given the use of flow cytometry in figure 1 to assess YAP levels. Underlining this is the fact that, for the western blot used to confirm the activity of the shRNA, the differing levels of GAPDH make comparison of the YAP levels in the control versus YAP-shRNA samples difficult to assess. As noted for figure 1, it's difficult to discern how many times each experiment has been done and also what "n=3 per group" actually means.

Figure 3. As noted above, this figure relies heavily on the veracity of flow cytometry analysis. And again, evidence that cells selected as positive versus negative needs to be provided. And also, the meaning of the n=3 per group needs to be clarified.

Figure 4. The data connected to this figure is far more compelling than that presented in the previous figures. First, the shift in YAP expression, measured by flow cytometry is definitive and, within he

same experiment, an appropriate control (lacZ) is included. This increases confidence in the downstream results, with respect to Ki67, NKX6.1 and c-peptid.

Figure 5. This figure investigates the relationship between modulating YAP signalling on the functionality of SC-beta cells. The figure itself segregates the activity of the over-expression vector and the YAP inhibitor. This data should have been combined in a figure that included treatment of the cells over expressing YAP with the YAP inhibitor. Such an arm would have confirmed the specificity of the YAP inhibitor as well as greatly strengthened the overall results.

Figure 6 explores the role of YAP modulation on the latter stages of pancreatic differentiation. Two main parameters are examined. SOX9 expression - using intracellular flow cytometry and beta cell functionality following transplantation. The same issues relating to the positioning of flow cytometry gates, the authenticity of the gates and the number of independent experimental repeats applies to the data surrounding assessment of SOX9 expression (Panels a-h). In addition, these experiments would have been far more convincing had the cells over-expressing YAP been included - not just because of the negative affect on functionality, but because this population may have helped with placement of flow cytometry gates. The remaining panels in this figure examine the function and structure of grafted material that had either been treated or not treated with verteporfin. It is not clear from this analysis of grafted cells whether verteporfin treatment did anything, either positive or negative to the final results. Certainly the analysis of the frequency of SOX9+ cells in the grafts suggests that verteporfin reduced the frequency of this population - although it is unclear what the denominator was for this calculation.

Summary: As will be evident from the above, this study has a number of technical issues which make interpretation of the results difficult. These issues relate to the inclusion of validation experiments that confirm that what is being measured is actually reflective of the true situation. The study includes 3 modulators of YAP signalling - the verteporin, shRNAs and over-expression constructs. It is very unfortunate that these three modalities are not carried through as a cohort in each of the experiments, enabling the validity of each to be measured against its counterpoint. Last, it's very difficult to understand how many times each experiment has been done - indeed, whether the study contains any independent experimental examples. This opacity is compounded by the statement in the methods section, statistics - "For all the experiments included in this study, at least three replicates were considered for the analysis using stage-matched controls as a reference." - which, as far as I can tell, is slightly opaque with regard to experimental repeats. Finally, I'm not entirely convinced that those in the field will find this study of general interest, and therefore it's suitability for Nature Communications needs to be questioned.

Response to referees' comments

We thank the reviewers for their comments, especially as many of the points they raised were quite constructive. We have added new experiments as requested and believe this has significantly improved the manuscript. What follows is a detailed schedule of our response to each one of the reviewers' comments. Overall, the reviewers appear to have accepted the main conclusions of the paper, but asked for more quantification of the results. In the light of their requests, we've added new results, repeated experiments and done quantification by flow cytometry in many instances, as detailed below. Furthermore, as suggested, we include additional data with other pluripotent stem cell lines to support our conclusions.

The reviewers' comments have been copied below and are underlined and in italic font. Our point-by-point responses follow directly below each comment.

Reviewer #1

In this manuscript, Rosado-Olivieri et al show that YAP, a downstream effector of the Hippo signalling pathway, restrains pancreatic endocrine progenitor differentiation, and that inhibition can be exploited to enhance in vitro generation of beta cells. It also confirms earlier work showing that YAP promotes the expansion of pancreatic multipotent progenitors. Interestingly, it provides evidence to support that the effect of YAP is specific, inasmuch as it is not mimicked by other maneuvers to block proliferation. This is an important study that provides proof-of-principle evidence to implicate a new pathway in efforts to differentiate beta cells from iPSC. Some aspects should be addressed for publication:

Major points:

1. Fig. 1. Many YAP+ cells appear to show cytoplasmic, non-nuclear patterns. This is not indicative of active YAP, and should be assessed in a more quantitative manner.

We assessed cytoplasmic and nuclear YAP expression in both NKX6.1+ and NGN3+ cells, before and during endocrine differentiation. The results are highlighted in Figure S1. We find that although both cytoplasmic and nuclear expression can be identified in multiple potent pancreatic progenitors (MPPs) at stage 4, YAP expression significantly declines during subsequent stages of differentiation. We described these data in the main text:

“Although both cytoplasmic and nuclear YAP expression is present in NKX6.1+ cells at stage 4, its expression in this subpopulation of MPPs further declines as differentiation proceeds into the endocrine lineage (Fig. 1f-g and S1c).”

and

“Immunostaining of cell clusters at this time point (stage 5, day 3) shows that nuclear and cytoplasmic expression of YAP in approximately 30% of all early endocrine progenitors (NGN3+; Fig. 1g-h)...”

2. It is difficult to assess the downregulation between stages 4 (NKX6.1 late PP) and stage 5 (endocrine progenitors) based on representative immunofluorescence images without more quantitative methods such as flow cytometry.

Following this helpful suggestion, we performed a time-course flow cytometry analysis of YAP co-expression in NKX6.1, CHGA+ endocrine and non-endocrine cells from three biological replicates. We detect a significant decrease in YAP expression in NKX6.1+ progenitor cells as they commit to the endocrine lineage. More importantly, we show that, although 80-85% of non-endocrine cells express YAP, endocrine cells lose YAP expression as they differentiate. At the end of the differentiation protocol, ~95% of all endocrine cells do not express this factor. These results are presented in Figure 1 and the following text was added to describe the results:

“Because differentiation of MPPs is not synchronous, both early (NGN3+) and late (CHGA+) endocrine progenitors are present at stage 5, day 3. Immunostaining of cell clusters at this time point (stage 5, day 3) shows that nuclear and cytoplasmic expression of YAP in approximately 30% of all early endocrine progenitors (NGN3+; Fig. S1b, d) whereas a high proportion of late endocrine cells (CHGA+) lose YAP expression (Fig. 1d-e h and S1b). The expression pattern of YAP is suggestive of a sustained downregulation of YAP in MPPs as they are specified as endocrine. Reduced YAP expression in endocrine cells persists upon completion of the directed differentiation

protocol (Fig. 1f): approximately 95% of insulin-expressing β and ChgA+ endocrine cells do not express YAP (Fig. 1e, h). At these latter stages of differentiation, YAP expression is largely restricted to non-endocrine cells that co-express the ductal marker SOX9+ (Fig. 1i and S1e-f).”

3. Fig. 2. Is the increase in ChgA+ cells observed with both shRNAs and chemical inhibition? What is the nature of these CHGA+ NKX6-1+ cells, are they glucagon + cells? Unless this is clarified it might make sense to either add a qualifying statement or remove these findings from this Figure.

The increase in ChgA+ cells is observed with both shRNAs and chemical inhibition (Figure 2 and 3). CHGA+/NKX6.1+ during endocrine differentiation are NKX6.1+ MPPs differentiating into the endocrine lineage. Most cells co-expressing these markers are either monohormonal SC- β cells or other endocrine cells that do not express glucagon. NKX6.1+ is absent in SC-glucagon+ cells (data not shown). We added the following statement to the results section:

“The enhanced differentiation of NGN3+ endocrine precursors with verteporfin produces an increase in CHGA+/NKX6-1+ endocrine progenitor cells by the end of stage 5. This is not observed with roscovitine treatment (Fig. S2c-f, assayed at stage 5, day 7).”

4. Figure 2, and other figures: Y axes should start at zero to avoid giving the impression that effects are larger than they really are.

We modified all figures so that Y axes start at zero.

5. The authors state that YAP inhibition with verteporfin leads to a significant decrease in the number of NKX6.1 progenitors. Can the authors confirm that the 28% reduction is the average reduction of the 3 replicates used and provide a measure of variation and statistical significance? Same question for Figure 3g and 40,4% increase in C-peptide/NKX6.1+ cells. Numbers in the text do not seem to match Figure 3g. Same concern for 17% reduction in Sox9+ cell reduction in Figure 6.

These values represent the average percent reduction or increase relative to control replicates. We modified the text to clarify this issue (Figure S4). In all cases, we performed unpaired two-side t-test as described in the text:

“Statistical analysis was performed using unpaired two-sided t tests unless stated otherwise. For all the experiments included in this study, 3 or more biological replicates were performed using stage-matched controls as a reference and confirmed with multiple independent experimental repeats.”

6. Fig. 4h. This seems to show number of C-peptide + cells/Ki 67+ (with Ki67 as the denominator). Since the number of C peptide+ cells increases it seems more appropriate to show Ki67+/C-peptide+ cells.

We modified this figure as suggested and present the data as Ki67+/C-peptide+ cells. We also included data for C-peptide- cells.

7. Figure 6. It is unclear if authors normalized the number of INS+ transplanted. Either way this should be clarified. If the increase in differentiation of endocrine cells is offset by a decrease in SOX9+ progenitors, the next question is what is the effect of verteporfin on the hypothetical pool of cells that are neither bona-fide multipotent progenitors or endocrine cells?

We did not normalize the number of INS+ transplanted cells. We injected 5×10^6 differentiated cells (per animal) collected at the end of the SC- β cell differentiation protocol. We expect that the effect of verteporfin is primarily on MPPs and non-endocrine cells. As most YAP+ cells at this stage co-express SOX9, we expect that this is the major subpopulation being affected by the treatment. The following statements have been added to clarify these points:

“In brief, 5×10^6 differentiated cells (per animal), collected at the end of the SC- β cell differentiation protocol, were dispersed using Accutase and resuspended in 200 μ L of RPMI1640 medium and kept on ice for 5 to 10 min before loading into a catheter for cell delivery under the kidney capsule.” (Methods)

“At these latter stages of the differentiation protocol, YAP expression is largely restricted to non-endocrine cells that co-express the ductal marker SOX9+ (Fig. 1i and S1e-f).” (Results)

8.- The discussion should perhaps clarify the authors view on whether

The original comment was incomplete when we received the responses and we did not know how to respond to it.

Minor comments

9. Figure 4b Minor figure editing is needed: Y-axis labels should be enlarged. Currently unreadable

We corrected the Y-axis in Figure 4b.

10. Stars representing statistical significance in Figure 5a and 6n as well as individual data points from Figure 5c should be enlarged.

Stars denoting statistical significance and data points were enlarged.

11. Calculation of stimulation index from GSIS assays is not described in Methods.

We have added a description of how we calculated stimulation indexes in Methods:

“Insulin secretion levels were normalized to total insulin content for each sample. Stimulation indexes were calculated as a ratio of insulin secretion at high glucose (20 mM) relative to the basal secretion (2.8 mM glucose).”

Reviewer #2

The data in this paper support an important but incremental advance in our knowledge of beta cell differentiation and development. YAP inhibition promotes increased generation of pancreatic progenitors, with some possible impact on beta cell function. See below for specific comments and concerns.

12. Figure 1: I am not sure that IF is the best way to characterize YAP expression in the different stages and populations of beta cell differentiation. Is this image representative of what is going on more broadly? I would like to see either FACS analysis or mRNA expression in sorted cell populations in addition to IF: this is shown only for stage 5 cells. A more comprehensive time course is necessary, especially as this is the basis for the hypothesis driving the rest of the paper. All FACS data should be aggregated from multiple experiments and analyzed with the appropriate statistical test to demonstrate any difference is real. Figure 1G is certainly unconvincing as a FACS plot of YAP positive versus YAP negative cells; more controls justifying this gating are needed.

Thank you for this suggestion. We performed a time-course flow cytometry analysis of YAP co-expression in NKX6.1, CHGA+ endocrine and non-endocrine cells from three biological replicates. We detect a significant decrease in YAP expression in NKX6.1+ progenitor cells as they commit to the endocrine lineage. More importantly, we show that, although 80-85% of non-endocrine cells express YAP, endocrine cells lose YAP expression as they differentiate. At the end of the differentiation protocol, around 95% of all endocrine cells do not express this factor. These results are presented in Figure 1 and the following statements were added to describe these results:

“Because differentiation of MPPs is not synchronous, both early (NGN3+) and late (CHGA+) endocrine progenitors are present at stage 5, day 3. Immunostaining of cell clusters at this time point (stage 5, day 3) shows that nuclear and cytoplasmic expression of YAP in approximately 30% of all early endocrine progenitors (NGN3+; Fig. S1b, d) whereas a high proportion of late endocrine cells (CHGA+) lose YAP expression (Fig. 1d-e h and S1b). The expression pattern of YAP is suggestive of a sustained downregulation of YAP in MPPs as they are specified as endocrine. Reduced YAP expression in endocrine cells persists upon completion of the directed differentiation protocol (Fig.1f): approximately 95% of insulin-expressing β and ChgA+ endocrine cells do not express YAP (Fig.

1e, h). At these latter stages of differentiation, YAP expression is largely restricted to non-endocrine cells that co-express the ductal marker SOX9+ (Fig. 1i and S1e-f).”

Figure IG was removed and replaced with quantification of YAP expression in NKX6.1+, CHGA+, and CHGA-cells during endocrine differentiation. We now include a description and example of the gating strategy in the Methods and Figure S3.

13. Figure 2J: the loading controls are extremely different, making evaluation of YAP protein knockdown problematic. The bands for the YAP protein blot do not look side-by-side because the backgrounds are different, whereas the control bands do appear side-by-side.

We modified this figure to side-by-side protein blots for YAP and GAPDH expression. These results were validated by qPCR and are also included in the new Figure 2.

14. Figure 3 A-D. Flow cytometry for NGN3 suggest an increase of NGN3+ population. Can this finding be confirmed by RNA or western blot.

-Figures 3E-G: the quantification in G is more impressive looking than the FACS plots in E and F would suggest. I would like to see more FACS plots in supplemental figures with appropriate controls to visually support the quantification.

- Figure 3E-G and 3I-K: why is the yield of C-peptide and Nkx6.1 double positive cells so different in controls between experiments?

- Use of cell cycle inhibitor roscovitine (Figure S2) was a good parallel experiment, but I would like to see more replicates as there did appear to be a trend towards a difference: an n of 3 may be underpowered.

- In addition the absolute yield of c-peptide+NKx6.1+ cells increased due to YAP inhibition should be shown, not just percentage.

We confirmed the results on NGN3 expression by qPCR and include the data in Figure S2a.

We now include data for staining controls and additional replicates in Figure S3 that support the quantification in Figure 3G.

These experiments were performed with two different batches of hPSC cells (HUES8 cells). There can be batch-to-batch variation in differentiation efficiencies. Thus, we performed the differentiations in parallel with hPSC cells obtained from the same maintenance batch as described in the Methods:

“Control and experimental samples were collected from differentiations performed in parallel with hPSC cells obtained from the same maintenance batch.”

We have included additional data for experiments performed with the cell cycle inhibitor, roscovitine. Cultures of progenitors supplemented with the inhibitor did not display differences in the proportion of NGN3+ cells or expression of this marker, as assayed by flow cytometry and qPCR, respectively. More importantly, it did not enhance the number of CHGA+NKX6.1+ progenitor cells compared to controls. The data is now presented in Figure S2 and described as below:

“Treatment with roscovitine, a cell cycle inhibitor, at this stage did not have the same effect (Fig. S2a-f), suggesting that blocking proliferation per se does not enhance differentiation. The enhanced differentiation of NGN3+ endocrine precursors with verteporfin produces an increase in CHGA+/NKX6-1+ endocrine progenitor cells by the end of stage 5, which is not observed with roscovitine treatment (Fig. S2c-f, assayed at stage 5, day 7).”

We do not see significant differences in total cell numbers at the end of the differentiation. Thus, we conclude that the increased proportion of C-peptide+NKX6.1+ upon completion of the experiment reflects an increase in absolute yield.

15. Figure 4F-G: Ki67 expression seems to vary along a continuum rather than be a bimodal process, which is interesting and not addressed. Has this previously been established? I would like to see more FACS controls in

supplemental figures to support and follow-up on this result. Cell yields and markers of apoptosis would be helpful to see what exactly YAP is doing. Ki67 staining does not necessarily equal proliferation. -In addition, an inducible overexpression of the stabilized YAP would be very helpful, as it could demonstrate if 1- can this method be used to meaningfully expand ES derived beta cells or progenitors, and 2- do these expanded cells generate functional beta cells once the YAP overexpression is stopped. Possibly adding the inhibitor to their OE cells may be a method to do this if the drug is still functional on the stabilized YAP.

Given that proliferation of cells at this stage is not synchronous, we expect to see variation in the expression levels of Ki67. Our gating of Ki67+ cells was supported by secondary-only staining controls as described in the Methods section. Moreover, we validated this phenotype by performing EdU staining of LacZ and YAPS6A-overexpressing cells that also shows an increase in proliferation levels of both SC- β and non- β cells. We have now included two orthogonal assays, EdU incorporation and Ki67 staining, to support our conclusion. Similar proportions of apoptotic cells were observed for LacZ and YAPS6A which ruled out apoptosis as a driver of the phenotypes reported here. These results are now presented in Figure 4 and the text is revised as follows:

“Consistent with a role of YAP expression in limiting endocrine differentiation, YAPS6A overexpression leads to a significant decrease in the proportion of C-peptide+/NKX6.1+ β cells, compared to controls, with no effect on cell apoptosis (Fig. 4c-e, l-m; stage 6 day 14). Notably, verteporfin treatment rescued the impaired endocrine differentiation observed upon YAPS6A overexpression (Fig. S6a-c). We quantified levels of proliferation of YAPS6A-overexpressing SC- β cells and detected a 5-fold increase in the proportion of proliferating cells, compared to controls, as measured by co-expression of the proliferation marker Ki67 with C-peptide and EdU incorporation in SC- β cells (Fig. 4f-j). This is consistent with previous reports²⁹.”

Thank you for the suggestion of using an inducible overexpression method. We now include experiments using a doxycycline-inducible system demonstrating that a transient overexpression of YAPS6a leads to increased proliferation of SC- β cells that become functional once overexpression is stopped. The data are presented in Figure S6 and described in text.

“Interestingly, a transient activation of doxycycline-inducible YAPS6A during the first 4 days of stage 6 leads to an increased proliferation of SC- β cells (Fig. S6d-f) but does not affect the function of SC- β cells when assayed at stage 6 day 12 (Fig.S6g-h).” (Methods)

16. Y axes on Figure 5A and 5C are 10-fold different, raising concerns about normalization. I would like to see the raw numbers (level of insulin secretion and total insulin content) graphed independently in supplemental figures to support the conclusions. Insulin/cpeptide secreted per insulin+ cell is another useful way to show these sorts of data. It looks that the inhibitor treated cells just secrete more insulin in all circumstances but SI does not seem to be impacted? There is no indication of a statistical test being performed in Figure 5b.

In Figure 5a and 5c, the axis scales were discordant due to our miscalculation during normalization. We have recalculated the data and the updated plots are shown in Figure 5a and 5c. We have also included the raw insulin secretion values and total insulin content in Fig. S5. Based on these data, the levels of insulin secretion of verteporfin-treated cells are higher than control cells. However, there was no difference in stimulation indexes. Results of statistical test are now included in the figure.

17- While GCG was examined in the supplemental materials, what about SST expression upon modulation of YAP pathway during differentiation? Considering that inhibitor treated cultures secrete more insulin in general, loss of SST could be a possible mechanism for this finding.

We did not detect an increase in the proportion of SST+ delta cells upon YAP inhibition and the results are now included in Figure S3j. Thus, loss of delta cells as a mechanism is unlikely.

18. The authors state they use two different genetic backgrounds but they only showed data from one cell line and it was not clear if the results are similar in both genetic backgrounds. Data from the second genetic background must be presented in supplemental materials for all major findings both in differentiation and function, with statistical tests performed to demonstrate that these effects are reproducible. Honestly, a third line would be very helpful for the most critical experiments such as drug inhibition on differentiation efficiency and beta cell function. The beta

cell differentiation field has had issues with protocol reproducibility and it is critical to demonstrate these effects work in multiple genetic backgrounds.

We include data for two other iPSC cell lines, 1016 and 13B, showing the effects of verteporfin on improved endocrine and SC- β cell differentiation (Fig. S4). Thus, the enhanced generation of SC-endocrine and β cell differentiation is not dependent on the cell line. We included the following statements to describe this result:

“More importantly, the effect of verteporfin on SC-endocrine and β cell differentiation is robust and independent of the genetic background of the hPSC cell line (Fig. S4).”

Minor issues

19- I dislike the frequent (but not ubiquitous) use of non-0 y axis origins to make data look more significantly different, though I suppose passing the statistical test is more important than any graph.

We modified all figures so that Y axes start at zero.

20.- Overall, it is not always clear from the legends what stage/day cells are being analyzed and there seem to be internal inconsistencies in yields or cell numbers from experiment to experiment.

We have attended to this issue and information about the stage and day the cells were collected is now presented in the figure legends and text.

21- Though verteporfin is an established inhibitor of YAP, I would have liked to see at least one control (FACS plot or Western blot) showing this.

In Figure S6, we include evidence on the specificity of verteporfin on YAP activity. Treatment with verteporfin rescued the impaired endocrine differentiation observed upon YAPS6A overexpression.

22. It is unclear if YAP's promotion of cell proliferation is a process specific to the beta cell or pancreatic progenitor or endoderm or is even a general process, especially given the conclusions of figure 6. This may not be able to be addressed experimentally (or may be beyond the scope of this paper), but warrants further discussion.

YAP's promotion of cell proliferation is not specific to beta cells and this is now explicitly stated in the revised manuscript. Our results agree with other studies suggesting that YAP induces the proliferation of SC- β and progenitor cells. The following statement was added to the Results:

“The increase in proliferation was not specific to C-peptide+ β cells as we observed an increase in the proliferation of non- β cells as well (Fig. 4i, k).”

23. - In figure 2 H-N, shRNA experiments do not show virus infection efficiency. It is really hard to conclude that shRNA-YAP drives to strong phenotype in NKX6.1+ population and proliferation without knowing what percentage of cells are carrying the virus.

Approximately 30-40% of progenitor cells express the shRNAs, as estimated by the percentage of GFP+ cells. This is now stated in Methods.

“For shRNA-infected samples, only infected GFP+ cells were included in the analysis (approximately in 30-40% of all cells).”

24.- Y axis labelling is confusing in figure 5b. Why is the dotted line (presumed baseline) on 1.5 SI. This is generally shown at 1.

We modified the labelling and placed the dotted line on baseline (y=1).

Reviewer #3

This manuscript explores the proposition that the TAZ/YAP pathway plays a role in pancreatic differentiation in vitro from pluripotent stem cells. During mouse pancreatic development, other studies have suggested that this is so, and therefore, there is some currency in the idea that manipulation of this pathway in vitro, in the right circumstances, could enhance the yield or quality of beta cells so derived.

25. Figure 1 Outlines the general differentiation protocol and performs a basic characterisation of this story's key protagonists over the course of differentiation. Attention is focussed on NGN3, YAP and NKX6.1. As will be unveiled in later figures, the relationship between YAP and NGN3 expression is of particular interest, with the potential observation that YAP expression is enriched in NGN3+ cells. This relationship is key to much of what follows, elevating the importance of this figure. The strength of the idea is based on immunofluorescence staining and intracellular flow cytometry, the latter of which can be a technique fraught with traps - particularly with poorly characterised antibodies or antiserum. Close examination of the image of cells at the endocrine progenitor stage indicates that much of the NGN3 staining is cytoplasmic. As far as I'm aware, cytoplasmic NGN3 is not commonly observed - and so, I believe it is incumbent upon the authors to show some additional validation that this staining is real. Turning to the intracellular flow cytometry, it's not clear what rationale was used to place the gates in their positions. Some explanation of this is required. Given that the primary antibody in this experiment is in fact a polyclonal sera, it would be very useful, necessary even, for the authors to include a non-specific control (say a sera directed against something else - that isn't expressed) to be confident about why and where these gates are set. This same argument applies to the YAP staining. In addition, I think it is necessary to show, probably using Q-PCR, that the proteins assayed by FACS are really enriched or otherwise in the various fractions. Until this is done, it's very difficult to unconditionally accept the data as it's presented. Parenthetically, the NGN3 staining in panel "i" looks far more convincing - but also, it does not look like 20%. Last, as with figures to follow, it's very difficult to figure out exactly how many times these experiments have been done - particularly with respect to the flow cytometry experiments.

The antibodies used in the study have been extensively used by other groups and validated. Importantly, the expression patterns observed with all the antibodies included in our paper display stage-specific expression, as assayed by immunostaining and flow cytometry. Nonetheless, additional validation was performed by using negative controls and control tissue samples. With respect to NGN3 staining, its expression is predominantly, but not exclusively, nuclear and highly specific to stage 5 endocrine progenitors (Figure 1d and S1b). Expression of NGN3 in cells from other stages is negligible.

For the flow cytometry analysis included in this study, a description and an example of the gating strategy is included in the revised manuscript in Fig. S3 and Methods:

“Samples stained only with secondary antibodies were used to gate on cells that are negative and accurately identified cells expressing the markers included in the analysis.”

We performed a time-course flow cytometry analysis of YAP co-expression in NKX6.1, CHGA+ endocrine and non-endocrine cells from three biological replicates. We detect a significant decrease in YAP expression in NKX6.1+ progenitor cells as they commit to the endocrine lineage. More importantly, we show that, although 80-85% of non-endocrine cells express YAP, endocrine cells lose YAP expression as they differentiate. At the end of the differentiation protocol, around 95% of all endocrine cells do not express this factor. These results are presented in Figure 1 and the following statements were added to describe these results:

“Because differentiation of MPPs is not synchronous, both early (NGN3+) and late (CHGA+) endocrine progenitors are present at stage 5, day 3. Immunostaining of cell clusters at this time point (stage 5, day 3) shows that nuclear and cytoplasmic expression of YAP in approximately 30% of all early endocrine progenitors (NGN3+; Fig. S1b, d) whereas a high proportion of late endocrine cells (CHGA+) lose YAP expression (Fig. 1d-e h and S1b). The expression pattern of YAP is suggestive of a sustained downregulation of YAP in MPPs as they are specified as endocrine. Reduced YAP expression in endocrine cells persists upon completion of the directed differentiation protocol (Fig. 1f): approximately 95% of insulin-expressing β and ChgA+ endocrine cells do not express YAP (Fig. 1e, h). At these latter stages of differentiation, YAP expression is largely restricted to non-endocrine cells that co-express the ductal marker SOX9+ (Fig. 1i and S1e-f).”

26. Figure 2 looks at the potential of YAP using a chemical inhibitor, verteporfin. Again, the position of FACS gates is something that needs to be addressed. At this point the number of experimental repeats also needs to be explicitly stated. The figure legend says n=3 per group - but in itself, this statement conveys insufficient information. This data should probably be presented as the mean and SEM for 3 independent experiments. If this is not the case then this needs to be stated. Similarly as for figure 1, there needs to be some independent confirmation that the FACS is reading out reality - something that could be accomplished using Q-PCR on sorted fractions. The other clear deficit with this figure is the absence of FACS showing the reduction YAP expression following introduction of shRNAs - this is so obvious given the use of flow cytometry in figure 1 to assess YAP levels. Underlining this is the fact that, for the western blot used to confirm the activity of the shRNA, the differing levels of GAPDH make comparison of the YAP levels in the control versus YAP-shRNA samples difficult to assess. As noted for figure 1, it's difficult to discern how many times each experiment has been done and also what "n=3 per group" actually means.

We now include a description and example of the gating strategy in Methods and Fig S3:

“Samples stained only with secondary antibodies were used to gate on cells that are negative and accurately identified cells expressing the markers included in the analysis.”

We include a description of the number of replicates in both figure legends and methods sections. And the data in most figures is presented as mean and SEM for 3 biological replicates. The text has been expanded in Methods to clarify the statistical analysis:

“Statistical analysis was performed by using unpaired two-sided t tests unless stated otherwise. For all the experiments included in this study, 3 or more biological replicates were considered for the analysis using stage-matched controls as a reference and confirmed with multiple independent experimental repeats.”

To validate the increased NGN3 expression upon YAP inhibition or downregulation, we include qPCR and Nanostring data (Fig. 3h and S2a).

For the validation of YAP shRNAs, we modified Figure 2 and included side-by-side protein blots for YAP and GAPDH expression. These results were validated by qPCR and are also included in Figure 2.

27. Figure 3. As noted above, this figure relies heavily on the veracity of flow cytometry analysis. And again, evidence that cells selected as positive versus negative needs to be provided. And also, the meaning of the n=3 per group needs to be clarified.

Please see response to similar concerns in the comment on Figure 2 (point 26 above).

28. Figure 4. The data connected to this figure is far more compelling than that presented in the previous figures. First, the shift in YAP expression, measured by flow cytometry is definitive and, within the same experiment, an appropriate control (lacZ) is included. This increases confidence in the downstream results, with respect to Ki67, NKX6.1 and c-peptide.

We validated the phenotype by performing EdU staining of LacZ and YAPS6A-overexpressing cells. With this assay, we show an increase in proliferation levels of both SC- β and non- β cells upon YAPS6A overexpression (Figure 4). We now include two assays, EdU incorporation and Ki67 staining, to support the conclusion. Similar proportions of apoptotic cells were observed for LacZ and YAPS6A ruling out apoptosis as the driver of the phenotype. We include the following statements to describe these experiments:

“Consistent with a role of YAP expression in limiting endocrine differentiation, YAPS6A overexpression leads to a significant decrease in the proportion of C-peptide+/NKX6.1+ β cells compared to controls with no effect on cell apoptosis (Fig. 4c-e, l-m; stage 6 day 14). Importantly, verteporfin treatment rescued the impaired endocrine differentiation observed upon YAPS6A overexpression (Fig. S6a-c). We quantified the levels of proliferation of YAPS6A-overexpressing SC- β cells and detected a 5-fold increase in the proportion of proliferating cells compared to control samples, as measured by co-expression of the proliferation marker Ki67 and EdU incorporation in SC- β cells (Fig. 4f-j), consistent with previous studies²⁸⁻²⁹.”

29. Figure 5. This figure investigates the relationship between modulating YAP signalling on the functionality of SC-beta cells. The figure itself segregates the activity of the over-expression vector and the YAP inhibitor. This data should have been combined in a figure that included treatment of the cells over expressing YAP with the YAP inhibitor. Such an arm would have confirmed the specificity of the YAP inhibitor as well as greatly strengthened the overall results.

We now include experiments using a doxycycline-inducible system for transient overexpression of YAPS6A. This leads to increased proliferation of SC- β cells that become functional once the overexpression is stopped. This is described in the manuscript:

“Interestingly, a transient activation of doxycycline-inducible YAPS6A during the first 4 days of stage 6, which leads to an increased proliferation of SC- β cells (Fig. S6d-f), did not affect the function of SC- β cells when assayed at stage 6, day 12 (Fig.S6g-h).” (Methods)

In Figure S6, we also include evidence to show the specificity of verteporfin on YAP activity. Treatment with verteporfin rescued the impaired endocrine differentiation observed upon YAPS6A overexpression.

30. Figure 6 explores the role of YAP modulation on the latter stages of pancreatic differentiation. Two main parameters are examined SOX9 expression - using intracellular flow cytometry and beta cell functionality following transplantation. The same issues relating to the positioning of flow cytometry gates, the authenticity of the gates and the number of independent experimental repeats applies to the data surrounding assessment of SOX9 expression (Panels a-h). In addition, these experiments would have been far more convincing had the cells over-expressing YAP been included - not just because of the negative affect on functionality, but because this population may have helped with placement of flow cytometry gates. The remaining panels in this figure examine the function and structure of grafted material that had either been treated or not treated with verteporfin. It is not clear from this analysis of grafted cells whether verteporfin treatment did anything, either positive or negative to the final results. Certainly the analysis of the frequency of SOX9+ cells in the grafts suggests that verteporfin reduced the frequency of this population - although it is unclear what the denominator was for this calculation.

We address concerns regarding the flow cytometry analysis and independent experimental repeats in the responses above.

Based on our analysis, there is a significant decrease in the frequency of Sox9+ cells of all grafted cells (total DAPI+). However, we did not find significant differences in graft function as both control and verteporfin-treated samples display robust *in vivo* GSIS responses.

Summary: As will be evident from the above, this study has a number of technical issues which make interpretation of the results difficult. These issues relate to the inclusion of validation experiments that confirm that what is being measured is actually reflective of the true situation. The study includes 3 modulators of YAP signalling - the verteporfin, shRNAs and over-expression constructs. It is very unfortunate that these three modalities are not carried through as a cohort in each of the experiments, enabling the validity of each to be measured against its counterpoint. Last, it's very difficult to understand how many times each experiment has been done - indeed, whether the study contains any independent experimental examples. This opacity is compounded by the statement in the methods section, statistics - "For all the experiments included in this study, at least three replicates were considered for the analysis using stage-matched controls as a reference." - which, as far as I can tell, is slightly opaque with regard to experimental repeats. Finally, I'm not entirely convinced that those in the field will find this study of general interest, and therefore it's suitability for Nature Communications needs to be questioned.

We have addressed all these issues in the responses above and in the revised manuscript. We thank you for your attention to these issues.

END

Reviewers' comments:

Reviewer #1 (Remarks to the Author):

The authors have addressed my suggestions and improved the manuscript.

Reviewer #2 (Remarks to the Author):

My concerns have been addressed

Reviewer #3 (Remarks to the Author):

It is not clear to me that the authors have really addressed any of the main concerns I raised in the original review of this manuscript. These centred on the reproducibility of data and its underlying authenticity - particularly regarding the validity of the gating strategies used for flow cytometry. I am a little concerned that key pieces of primary data in the original figure 1 have disappeared (Being the expression of NGN3 and CHGA as a function of YAP expression). These have been replaced by series of bar graphs and single FACS plot of SOX9 versus YAP. The actual controls for this plot are not shown - yet the quantity of FACS based data now present in figure 1 has increased. Furthermore, the prominence of the NGN3 in the overall story seems to have diminished.

If the authors wish to include any data relating to the level of YAP expression, they really need to present some convincing flow cytometry for YAP. Starting with YAP over-expression experiments and shRNA inhibition experiments. The authors need to show that, using flow cytometry, the levels of YAP change under conditions where the amount of YAP protein is supposed to go up and down. For all the experiments, the secondary only control is not really adequate - and does not account for the influence of non-specific sticking of the primary antibody and it's subsequent detection with the secondary. The statement "Samples stained only with secondary antibodies were used to gate on cells that are negative and accurately identified cells expressing the markers included in the analysis" does not constitute proof that this strategy worked. For each of the antibodies used in this analysis, the authors should sort the cells and determine the RNA content for the specific sorting parameter, an indication that the sorting strategy is indeed identifying cells enriched for specific populations.

The authors have also not really addressed my comment regarding the replication of the experiments - and specifically, my comment in relation to the sentence in the original manuscript "For all the experiments included in this study, at least three replicates were considered for the analysis using stage-matched controls as a reference." I am also a little uncomfortable about the meaning of 3 biological replicates. Does this mean the actual differentiation experiment was performed three times - and therefore each sample represents an analysis of an independent experimental set up - or does it mean that 3 samples were harvested from the same experiment? Could they please clarify.

Overall, I believe that most of the manipulations described in this manuscript did indeed affect the trajectory of differentiation as the authors conclude. My main concern however, relates to the interpretation of the mechanisms underpinning these changes. In particular, the relationship between YAP and NGN3 and differentiation. For this relationship to be believable, the strength of the flow cytometry data is critical.

Response to referees' comments

We thank all the reviewers for their continued attention to our manuscript. It seems that all 3 reviewers have accepted the main conclusions of our paper and the additional experiments we included in the revised manuscript. We now direct our attention to additional data supporting the flow cytometry-based quantification as requested by Reviewer #3 as well as clarifying some statements in the text.

The reviewer's comments are underlined below.

Reviewer #1

The authors have addressed my suggestions and improved the manuscript.

Reviewer #2

My concerns have been addressed.

Reviewer #3

It is not clear to me that the authors have really addressed any of the main concerns I raised in the original review of this manuscript. These centred on the reproducibility of data and its underlying authenticity - particularly regarding the validity of the gating strategies used for flow cytometry. I am a little concerned that key pieces of primary data in the original figure 1 have disappeared (Being the expression of NGN3 and CHGA as a function of YAP expression). These have been replaced by series of bar graphs and single FACS plot of SOX9 versus YAP. The actual controls for this plot are not shown - yet the quantity of FACS based data now present in figure 1 has increased. Furthermore, the prominence of the NGN3 in the overall story seems to have diminished.

We are sorry to learn that the revisions did not satisfy all your concerns as we tried to address each point raised. We have now added additional flow cytometry data in Supplementary Figure 1 to support the quantification presented in Figure 1. These data include controls as well as an analysis of YAP co-expression with NKX6.1 and CHGA during the differentiation of progenitors into the endocrine lineage as requested.

Additionally, in our revised manuscript, we now include an analysis of YAP expression in NGN3+ endocrine cells in Supplementary Figure 1e & 1g, replacing the flow cytometry-based quantification included in the original manuscript. Importantly, these new data did not alter the conclusion; the data support the original conclusion that YAP expression is downregulated as progenitor cells commit to the endocrine lineage. These results are described in the main text as detailed below:

“Because differentiation of MPPs is not synchronous, both early (NGN3+) and late (CHGA+) endocrine progenitors are present at stage 5, day 3. Immunostaining of cell clusters at this time point shows nuclear and cytoplasmic expression of YAP in approximately 30% of early endocrine progenitors (NGN3+; Fig. S1e, g) whereas a high proportion of late endocrine cells (CHGA+) have lost YAP expression (Fig. 1d-e, h and **S1c, e**).”

If the authors wish to include any data relating to the level of YAP expression, they really need to present

some convincing flow cytometry for YAP. Starting with YAP over-expression experiments and shRNA inhibition experiments. The authors need to show that, using flow cytometry, the levels of YAP change under conditions where the amount of YAP protein is supposed to go up and down. For all the experiments, the secondary only control is not really adequate - and does not account for the influence of non-specific sticking of the primary antibody and it's subsequent detection with the secondary. The statement "Samples stained only with secondary antibodies were used to gate on cells that are negative and accurately identified cells expressing the markers included in the analysis" does not constitute proof that this strategy worked. For each of the antibodies used in this analysis, the authors should sort the cells and determine the RNA content for the specific sorting parameter, an indication that the sorting strategy is indeed identifying cells enriched for specific populations.

We now include flow cytometry-based quantification of YAP expression in cultures of cells expressing YAP-targeted shRNAs or transfected with YAPS6A overexpression vectors in Figures 3l and 4c, respectively. These data provide evidence for an effective and significant downregulation of YAP upon shRNA expression and overexpression of this factor in cultures infected with expression vectors.

We believe that gating on secondary only staining accounts for background fluorescence and allows us to gate cells that are negative for marker expression. This is an approach or method employed by many of the studies in the literature, including references we cite in our manuscript using the very same antibodies. We also use single primary antibody staining controls to validate this gating strategy.

Although the suggestion of validating each of the antibodies by sorting and RNA analysis is sound, we do not believe it will not provide a secure conclusion because the expression of some of the markers we include in our analysis, including NGN3, SOX9 and YAP, are regulated post-transcriptionally. Moreover, the experiments requested for the validation of the antibodies are also technically very challenging as they require large quantities of fixed cells for sorting and an independent protocol optimization for RNA recovery and analysis post-fixation. We believe it unreasonable to require validation of each of these antibodies as they have been included and independently validated in many studies.

The authors have also not really addressed my comment regarding the replication of the experiments - and specifically, my comment in relation to the sentence in the original manuscript "For all the experiments included in this study, at least three replicates were considered for the analysis using stage-matched controls as a reference." I am also a little uncomfortable about the meaning of 3 biological replicates. Does this mean the actual differentiation experiment was performed three times - and therefore each sample represents an analysis of an independent experimental set up - or does it mean that 3 samples were harvested from the same experiment? Could they please clarify.

Our analysis included data from three independent differentiations. A description of how we performed the statistical analysis is in the Methods and is copied below:

"For all the experiments included in this study, 3 or more biological replicates were performed using stage-matched controls as a reference."

Overall, I believe that most of the manipulations described in this manuscript did indeed affect the trajectory of differentiation as the authors conclude. My main concern however, relates to the interpretation of the mechanisms underpinning these changes. In particular, the relationship between YAP and NGN3 and differentiation. For this relationship to be believable, the strength of the flow cytometry data is critical.

REVIEWERS' COMMENTS:

Reviewer #3 (Remarks to the Author):

The new data included in supplementary figure 1b,c provides convincing evidence that the flow cytometry quantification of YAP is accurate, particularly when coupled with the analysis of the shRNA inhibition and over-expression in shown in figures 3l and 4c. These were my main concerns and I believe the authors have adequately addressed them.

Response to reviewers' comments

REVIEWERS' COMMENTS:

Reviewer #3 (Remarks to the Author):

The new data included in supplementary figure 1b,c provides convincing evidence that the flow cytometry quantification of YAP is accurate, particularly when coupled with the analysis of the shRNA inhibition and over-expression in shown in figures 3l and 4c. These were my main concerns and I believe the authors have adequately addressed them.

We are glad we addressed your main concerns. Thank you for the feedback!